# TRACKING OBJECTS THAT CHANGE IN APPEARANCE WITH PHASE SYNCHRONY

**Sabine Muzellec**[*]
CerCo, CNRS, Universite de Toulouse, France
Carney Institute for Brain Science
Brown University, USA
sabine_muzellec@brown.edu

**Drew Linsley**[*]
Carney Institute for Brain Science
Department of Cognitive & Psychological Sciences
Brown University, USA
drew_linsley@brown.edu

**Alekh K. Ashok**
Carney Institute for Brain Science
Department of Cognitive
& Psychological Sciences
Brown University, USA

**Ennio Mingolla**
Northeastern University
Boston, MA, USA

**Girik Malik**
Northeastern University
Boston, MA, USA

**Rufin VanRullen**
CerCo, CNRS
Universite de Toulouse
France

**Thomas Serre**
Carney Institute for Brain Science
Department of Cognitive & Psychological Sciences
Brown University, USA

## ABSTRACT

Objects we encounter often change appearance as we interact with them. Changes in illumination (shadows), object pose, or the movement of non-rigid objects can drastically alter available image features. How do biological visual systems track objects as they change? One plausible mechanism involves attentional mechanisms for reasoning about the locations of objects independently of their appearances — a capability that prominent neuroscience theories have associated with computing through neural synchrony. Here, we describe a novel deep learning circuit that can learn to precisely control attention to features separately from their location in the world through neural synchrony: the complex-valued recurrent neural network (CV-RNN). Next, we compare object tracking in humans, the CV-RNN, and other deep neural networks (DNNs), using FeatureTracker: a large-scale challenge that asks observers to track objects as their locations and appearances change in precisely controlled ways. While humans effortlessly solved FeatureTracker, state-of-the-art DNNs did not. In contrast, our CV-RNN behaved similarly to humans on the challenge, providing a computational proof-of-concept for the role of phase synchronization as a neural substrate for tracking appearance-morphing objects as they move about.

## 1 INTRODUCTION

Think back to the last time you prepared a meal or built something. You could keep track of the objects around you even as they changed in shape, size, texture, and location. Higher biological visual systems have evolved to track objects using multiple visual strategies that enable object tracking under different visual conditions. For instance, when objects have distinct and consistent appearances over time, humans can solve the temporal correspondence problem of object tracking by "re-recognizing" them (Fig. 1a, (Pylyshyn & Storm, 1988; Pylyshyn, 2006)). When two or more objects in the world look similar to each other, and re-recognition becomes challenging, a complementary strategy is to track one of them by integrating their motion over time (Fig. 1b, (Lettvin et al., 1959; Takemura et al., 2013; Kim et al., 2014; Adelson & Bergen, 1985; Frye, 2015; Linsley et al., 2021)).

The neural substrates for tracking objects by re-recognition or motion integration have been the focus of extensive studies over the past half-century. The current consensus is that distinct neural circuits are

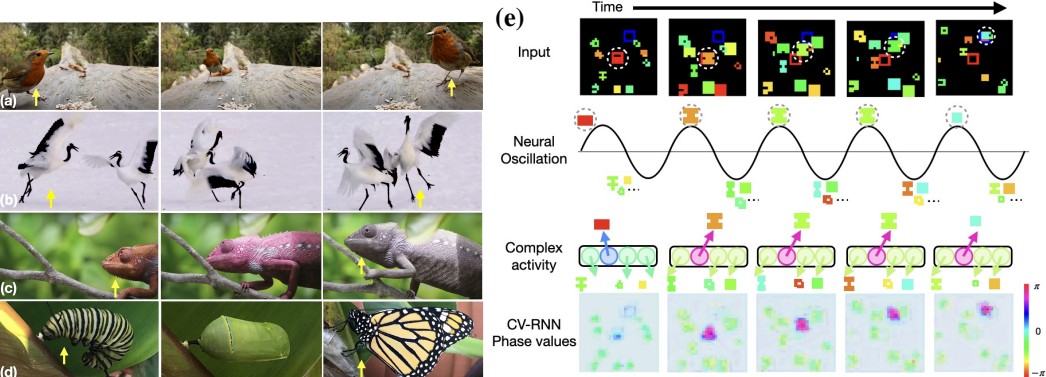

Figure 1: **How do Biological visual systems track the object tagged by the yellow arrow? (a)** Sometimes, the object's appearance makes it easy to track (Pylyshyn, 2006; Pylyshyn & Storm, 1988). **(b)** Other times, when objects look similar, the target can be tracked by following its motion through the world (Lettvin et al., 1959; Takemura et al., 2013; Kim et al., 2014; Adelson & Bergen, 1985; Frye, 2015; Linsley et al., 2021). Here, we investigate a computational problem that has received far less attention: how do biological visual systems track objects when their colors, textures **(c)**, or shapes **(d)** change over time? **(e)** We developed the `FeatureTracker` challenge to systematically evaluate humans and machine vision systems on this problem. In `FeatureTracker`, observers watch videos containing objects that change in color and/or shape over time, and have to decide if the target object, which begins in the red square (circled in white for clarity), ends up in the blue square by the end of a video. When presented with a `FeatureTracker` video, one possible strategy suggested by neuroscience theories is that the oscillatory activity of neural populations can keep track of different objects over time. Specifically, the target is encoded by a population of neurons that fire with a timing that differs from that of the population that responds to the distractors Astrand et al. (2020). We approximate the cycle of the oscillation with complex-valued neurons. In the CV-RNN, the phase of a complex-valued neuron represents the object encoded by this neuron. The CV-RNN thus learns to tag the target with a phase value different from the phase value of the distractors.

responsible for each strategy (Lettvin et al., 1959; Takemura et al., 2013; Kim et al., 2014; Adelson & Bergen, 1985; Frye, 2015; Pylyshyn, 2006). Much less progress has been made in characterizing how visual systems track objects as their appearances change (Fig. 1c,d). However, visual attention likely plays a critical role in tracking (Blaser et al., 2000). Visual attention is considered essential to solve many visual challenges that occur during object tracking, such as maintaining the location of an object even as it is occluded from view (Koch & Ullman, 1987; Roelfsema et al., 1998; Busch & VanRullen, 2010; Herrmann & Knight, 2001; Pylyshyn & Storm, 1988; Pylyshyn, 2006). We hypothesize that visual attention similarly helps when tracking objects that change appearance by maintaining information about their location in the world independently of their appearances.

How is this type of visual attention implemented in the brain? Prominent neuroscience theories have proposed that the synchronized firing of neurons reflects the allocation of visual attention. Specifically, neural synchrony enables populations of neurons to multiplex the appearance of objects with more complex visual routines controlled by attention (McLelland & VanRullen, 2016; Wutz et al., 2020; Frey et al., 2015). Neural synchrony could, therefore, help keep track of objects regardless of their exact appearance at any point in time.

Previous work proposed using complex-valued representations in RNNs (Lee et al., 2022), and in other architectures to implement neural synchrony in artificial models (Reichert & Serre, 2013; Löwe et al., 2022; Stanić et al., 2023). According to the framework proposed by Reichert & Serre (2013), each neuron in an artificial neural network can be represented as a complex number where the magnitude encodes for specific object features, and the phase groups the features of different objects. Such representations allow the modeling of various neuroscience theories (Singer & Gray, 2003; Singer, 2007; 2009) related to the role of neural synchrony. Here, we investigate whether the use of complex-valued representations to implement neural synchrony can help to solve the `FeatureTracker` challenge through large-scale computational experiments (see Fig. 1e).

**Contributions.**  The appearances of objects often change as they move through the world. To systematically measure the tolerance of observers to these changes, we introduce the `FeatureTracker` challenge: a synthetic tracking task where the motion, color, and shape of objects are precisely controlled over time (Fig. 1e). In each `FeatureTracker` video, a human observer or a machine vision algorithm has to decide if a target object winds up in a blue square after beginning in a red square. The challenge is made more difficult by the presence of non-target objects that also change in appearance over time and which inevitably cross paths with the target, forcing observers to solve the resulting occlusions (Pylyshyn & Storm, 1988; Blaser et al., 2000; Linsley et al., 2021). This challenge can be further modulated by training and testing observers on objects with different appearance statistics. Through a series of behavioral and computational experiments using `FeatureTracker`, we discover the following:

- Humans are exceptionally accurate at tracking objects in the `FeatureTracker` challenge as these objects move through the world and change in color, shape, or both.

- On the other hand, DNNs struggle on `FeatureTracker`, especially when object color spaces differ between training and test.

- Inspired by neuroscience theories on how populations of neurons implement solutions to the binding problem of `FeatureTracker`, we incorporated a novel mechanism for computing attention through neural synchrony, using complex-valued representations, in a recurrent neural network architecture, which we call the complex-valued recurrent neural network (CV-RNN). The CV-RNN approaches human performance and decision-making on `FeatureTracker`.

- Our findings establish a proof-of-concept that neural synchrony may support object tracking in humans, and can induce similar capabilities in artificial visual systems. We release `FeatureTracker` data, code, and human psychophysics at `https://github.com/S4b1n3/feature_tracker` to help the field investigate this gap between human and machine vision.

## 2  BACKGROUND AND RELATED WORK

**Visual routines**  Ullman (1984) theorized that humans can compose atomic attentional operations, like those for segmenting or comparing objects, into rich "visual routines" that support reasoning. He further proposed that the core set of computations that comprise visual routines can be flexibly reused and applied to objects regardless of their appearance, making them a strong candidate for explaining how humans can track objects that change in appearance. Visual routines are likely implemented in brains through feedback circuits that control attention (Roelfsema et al., 2000), and potentially through neural synchrony (McLelland & VanRullen, 2016). Developing a computational understanding of how visual routines contribute to object tracking and how they might be implemented in brains would significantly advance the current state of cognitive neuroscience.

**Computing through neural synchrony**  The empirical finding that alpha/beta (12–30Hz) and gamma (>30Hz) oscillations tend to be anti-correlated in primate cortex has motivated the development of theories on how the temporal synchronization of different groups of neurons may reflect an overarching computational strategy of brains. In the communication-through-coherence (CTC) theory, Fries (2015) proposed that alpha/beta activity carries top-down attentional signals, which reflect information about the current context and goals. Others have expanded on this theory to suggest that these top-down signals can be spatially localized in the cortex to multiplex attentional computations independently of the features encoded by neurons (Miller et al., 2024). While there have been many different theories proposed on how computing through oscillations works (McLelland & VanRullen, 2016; Lisman & Jensen, 2013; Grossberg, 1976; Milner, 1974; Mioche & Singer, 1989), here we assume an induced oscillation and study synchrony as a mechanism for visual routines and its potential for implementing object tracking in brains.

**Generalization and shortcut learning in DNNs**  A drawback of DNNs' great power is their tendency to learn spurious correlations between inputs and labels, which can lead to poor generalization (Barbu et al., 2019; Geirhos et al., 2020b). Moreover, while object classification models have grown more accurate over the past decade — now matching and sometimes exceeding human performance (Shankar et al., 2020) — they have done so by learning recognition strategies that are

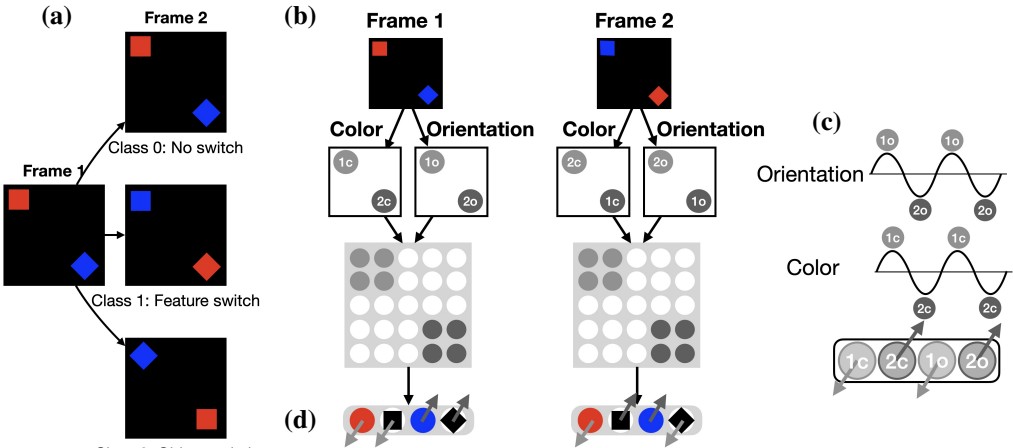

Figure 2: **Neural synchrony helps track objects that change in appearance. (a)** The `shell game` is designed to probe how a neural network, with the functional constraints of biological visual systems, could track objects as they change in appearance between frames one and two. Are the two images the same, or has the objects' color and/or orientation flipped (three possible responses)? **(b)** We tested a simplified model of the hierarchical visual system on the task, which consisted of two layers of neurons: (*i*) a convolutional layer with high-resolution feature maps, followed by (*ii*) a spatial average pooling of neuron responses and a layer of recurrently connected neurons (McLelland & VanRullen, 2016). 1c/2c are object colors, 1o/2o are object orientations; the loss of spatial resolution between the layers causes these object features to interfere. The model can detect the features present in the frame (red and blue color, as well as square and diamond orientations), but fails at binding the color and orientation with the position – hence cannot differentiate Frame 1 from Frame 2. **(c, d)** The same architecture can learn to solve the task with a complex-valued mechanism for neural synchrony, in which the magnitude of neurons captures object appearances, and the phase captures object locations.

becoming progressively less aligned with humans (Fel* et al., 2022; Linsley et al., 2023b;a). Synthetic datasets like `FeatureTracker` are useful for understanding why this misalignment occurs and guiding the development of novel architectures that can address it. The PathFinder challenge, which was originally developed to investigate the ability of observers to trace long curves in clutter (Linsley et al., 2018), was used to optimize Transformer and modern state space model architectures (Tay et al., 2021; Gu et al., 2021; Smith et al., 2022). The most similar challenge to our `FeatureTracker` is PathTracker, which tested whether observers could track one object in a swarm of identical-looking objects as they briefly occlude each other while they move around (Linsley et al., 2021). Here, we extend PathTracker by adding parametric control over the shape and color of objects to test tracking as object appearances smoothly change.

**Complex-valued representations in artificial neural networks.** The neural network architectures that have powered the deep learning revolution can be seen as modeling the rates of neurons instead of their moment-to-moment spikes. Given this constraint, there have been multiple attempts to introduce neural synchrony into these models by transforming their neurons from real- to complex-valued. Early attempts at this approach showed that object segmentation can emerge from the phase of these complex-valued neurons (Zemel et al., 1995; Weber & Wermter, 2005; Reichert & Serre, 2013; Behrmann et al., 1998). These models relied on shallow architectures, small and poorly controlled datasets, and older training routines like the energy-based optimization methods used in Boltzmann machines that have fallen out of favor over recent years. Recently, there has been a renewed interest in neural synchrony as a mechanism for DNNs (Löwe et al., 2022; Stanić et al., 2023). Unlike these previous attempts, our CV-RNN only uses synchrony with complex-valued representations in its attention module. This makes the model far more scalable than prior attempts, as complex-valued units are at least twice as expensive as real-valued ones (only certain levels of quantization are possible with the former), and enables its use with spatiotemporal data.

## 3 MOTIVATION

How do biological visual systems track objects while they move through the world and change in appearance? Given that this problem has received little attention until now, we began addressing it through a toy experiment. We developed a simple `shell game` where observers had to describe how the colors, locations, and shapes of two objects changed from one point in time to the next (Fig. 2a; see SI A.5.1 for additional details). We then created a highly simplified model of a hierarchical and recurrent biological visual system to identify any challenges it may face with this game. The model was composed of an initial convolutional layer with high-resolution spatial feature maps, followed by a global average pooling layer (to approximate the coarser representations found in inferotemporal cortex, and a layer of recurrent neurons with more features than the first layer but no spatial map (McLelland & VanRullen (2016), Fig.2b). The convolutional layer was implemented using a standard PyTorch Conv2D layer, whereas the recurrent layer was implemented with the recently developed Index-and-Track (InT) recurrent neural network (RNN), which includes an abstraction of biological circuits for object tracking (Linsley et al., 2021). The combined model was trained on a balanced dataset of $10,000$ samples from the `shell game` using a Cross-Entropy loss and the Adam optimizer (Kingma & Ba, 2014). In conditions of the game when the objects changed positions, this model performed close to chance ($45\%$ accuracy). The loss of spatial resolution between the model's early and deeper layers caused its representations of each object's appearance and location to interfere with each other (Figs. A.1 and A.2; see SI A.5 for more details).

**Neural synchrony can implement visual routines for object tracking**
The retinotopic organization of hierarchical visual systems provides an important constraint for developing models of object tracking: the spatial resolution of representations decreases as they move through the hierarchy. We need a mechanism that can resolve the interference that this loss of spatial resolution causes to object representations without expanding the capacity of the model. One potential solution to this problem is neural synchrony, which can multiplex different sources of information within the same neuronal population with minimal interference (Sternshein et al., 2011; Drew et al., 2009). Similarly, synchrony has been proposed to implement object-based attention to form perceptual groups based on their gestalt (Woelbern et al., 2002; Elliott & Müller, 2001). We, therefore, hypothesized that neural synchrony could rescue model performance in the `shell game` (Fig.2c).

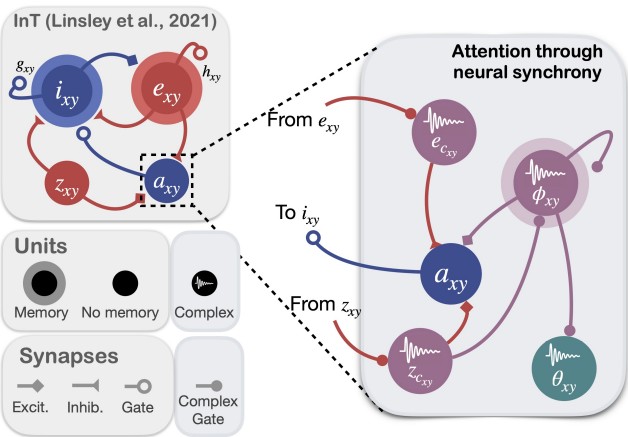

Figure 3: **Implementing neural synchrony through the complex-valued RNN (CV-RNN)**. The CV-RNN augments the InT RNN from Linsley et al. (2021) (shown on the left) with neural synchrony attention through the use of complex-valued units (shown on the right). In the CV-RNN, $e_c$ and $z_c$ convert $e$ and $z$ to the complex domain, $\phi$ is a recurrent unit maintaining a complex representation of the input, and $\theta$ transforms $\phi$ into a spatial map of the current frame.

We adapted the recurrent InT circuit used in the second layer of our simplified biological visual system model into a new neural architecture capable of learning neural synchrony using complex-valued representations. This recurrent neural network (RNN) (Linsley et al., 2021), inspired by neural circuit models of motion perception (Berzhanskaya et al., 2007) and executive cognitive function (Wong & Wang, 2006), contains an attention module that can learn to track objects by integrating their motion (see SI A.6.1 for details). We reasoned that augmenting this attention module with neural synchrony could help the entire model learn to solve the `shell game`. Specifically, complex-valued neurons could enable the attention module to bind object features by synchronizing the phase of its neurons encoding features sharing the same location, and desynchronizing the phases when the location differs.

The InT circuit consists of a feedforward drive $z \in \mathbb{R}$, an excitatory unit $e \in \mathbb{R}$, and an inhibitory unit $i \in \mathbb{R}$. The activities of these units are computed at column $x$, row $y$, and timestep $t$ of a spatiotemporal input through convolutions with the weight kernels $\mathbf{W}_a$ and $\mathbf{W}_z \in \mathbb{R}^{1,1,c,c}$, $\mathbf{W}_e$, and $\mathbf{W}_i \in \mathbb{R}^{5,5,c,c}$ with $c$ the hidden dimension of the circuit (here $c = 32$). An attention module computes activities $a$ that will eventually modulate $i$ in the following way:

$$a[t] = \sigma(\mathbf{W}_z * z[t] + \mathbf{W}_a * e[t-1]) \tag{1}$$

Here, $\sigma$ is a sigmoid pointwise nonlinearity, and $*$ is a convolution operator. We introduce neural synchrony into this attentional module by (*i*) transferring the real-valued activity ($z$ and $e$) to the complex domain, (*ii*) introducing a complex-valued recurrent unit $\phi[t]$, that stores the representation of each frame, and (*iii*) transferring its complex-valued output back to the real domain so that it can modulate $i$, as it normally does in the InT (Fig. 3). The resulting attentional module becomes:

$$a[t] = \sigma(In(||z_c[t] + \mathbf{W}_a^C * e[t] + \phi[t]||)) \quad \text{with} \quad z_c[t] = \mathbf{W}_z^C * z \text{ and } \phi[t] = \mathbf{W}_a * (z_c[t] + \phi[t-1]) \tag{2}$$

where $\mathbf{W}_z^C$ and $\mathbf{W}_a^C \in \mathbb{C}$ are complex weights that transfer the initial real-valued activity into the complex domain, $\mathbf{W}_a \in \mathbb{R}$ are real weights acting on complex-valued activity, and $In$ is an InstanceNorm (Ulyanov et al., 2016) operator that distributes the neural amplitudes (denoted by $||.||$) around 0 before applying the sigmoid function (see SI A.6.3 for more details).

**Solving the `shell game` with the CV-RNN.** We replaced the second layer of our simplified model of the visual cortex with the CV-RNN (Fig.2 and SI Table 3). We also introduced complex valued neurons into the model's first layer to ensure that the phases of layer 2 attentional neurons could capture the position information of objects. We reasoned that by training this model to solve the `shell game`, it could avoid the interference that affected the real-valued version by learning to use its neurons' magnitudes and phases to represent object features and locations separately. We confirmed our hypothesis: the CV-RNN perfectly solved the `shell game` (see SI A.6.4 and Fig. A.1 for details), implying a possible role of neural synchrony for tracking objects as they change appearance.

## 4 THE FEATURETRACKER CHALLENGE

**Overview** Our `shell game` was a highly simplified test of object tracking. In the real world, objects and their appearance and spatial features evolve smoothly and predictably over time. To better understand the capabilities of our CV-RNN — and neural synchrony — to track objects under such conditions, we next developed the `FeatureTracker` challenge. In `FeatureTracker`, observers watch a video and decide if a target object, which begins in a red square, travels to a blue square by the end of the video as opposed to a distractor (Fig. 4). This task is challenging for two reasons: (*i*) each video contains distractor objects that sometimes pass by and occlude the target object, and (*ii*) the color and/or shape of all objects in each video morph over time in precisely controlled ways.

**Design** Videos in the `FeatureTracker` challenge consists of 32 frames that are $32 \times 32$ pixels. Every frame shows a red start square, a blue goal square, a target object (defined as the object inside the red square at the start of the video), and 10 other "distractor" objects. As the objects move, their shapes, colors, or shapes and colors change in smooth and predictable ways (Fig.A.3; SI A.7.1 for details). Positive samples are when the target object ends in the blue square by the end of the video. In negative samples, a non-target object winds up in the blue square by the end of the video.

We created the `FeatureTracker` challenge to probe how well observers could track objects as their appearances changed in familiar or unfamiliar ways. We did this by sampling the starting state of each object's color and/or shape from distributions that we varied from training to test time. To elaborate, object appearances for training were sampled from one distribution (Fig. 4, red cube); then observers were tested on objects with appearances sampled in four different ways: (*i*) colors/shapes from the same distribution, (*ii*) colors from a different distribution but shapes from the same distribution, (*iii*) colors from the same distribution but shapes from a different distribution, or (*iv*) colors from a different distribution and shapes from a different distribution (Fig. 4 and SI A.7.2). We systematically evaluated the abilities of humans and machine vision systems to track objects with changing appearances by comparing their performances and decision strategies on each test set.

**Human benchmark** We began by evaluating humans on `FeatureTracker`. We recruited 50 individuals using Prolific to participate in this study. Participants viewed `FeatureTracker` videos

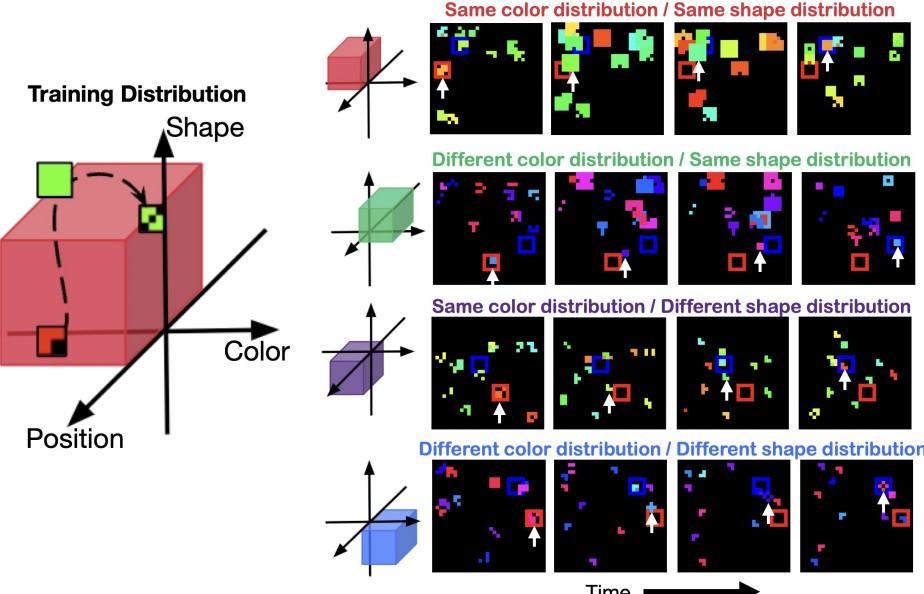

Figure 4: **The `FeatureTracker` challenge is a controllable environment where the objects can evolve along three feature dimensions: position, shape, and color.** The training distribution is generated from objects evolving in the upper-left quadrant of the 3D space (red cube), corresponding to half of the possible colors and shapes. The other testing conditions contain respectively objects of colors sampled from the other half of the spectrum but the same shapes (upper-right quadrant – green cube), same colors but different shapes (lower-left quadrant – purple cube), unseen colors and shapes (lower-right quadrant – blue cube). The task is to track the target located in the red marker in the first frame and to assess whether this target (shown here with a white arrow to improve visibility) or a distractor reaches the blue marker at the end of the video.

and pressed a button on their keyboard to indicate if the target object or a distractor reached the goal. Videos were played at $256 \times 256$ pixels with HTML5, which ensured consistent frame rates (Eberhardt et al., 2016). The experiment began with a 20-trial "training" stage (images from Fig. 4, red cube), which familiarized participants with the goal of `FeatureTracker` and how objects could change over time. Once training was completed, participants were tested on 120 videos. The experiment was not paced and lasted approximately 20 minutes. Participants were provided their informed consent before the experiment and were paid for their time. See SI. A.10 for an example and more details.

All participants viewed the same `FeatureTracker` videos to maximize the statistical power of our comparisons between the decision-making strategies of humans and machine vision systems. Participants were significantly above chance on all four `FeatureTracker` test sets ($p < 0.001$, test details and more statistics in SI A.10.2). Human performance also improved as more appearance cues changed. For example, humans were 92% accurate at tracking when the colors and shapes of objects changed but 79% accurate when both were fixed. These findings validate our assumption that humans are more than capable of tracking objects that change appearance.

**Results** Can the CV-RNN circuit and its version of neural synchrony match humans on `FeatureTracker`? To test this, we incorporated the circuit into an architecture that could be trained end-to-end to solve `FeatureTracker`. This architecture consisted of a convolutional layer with 32 $1 \times 1$ width filters, a 32-channel CV-RNN circuit that recurrently processed frames from each `FeatureTracker` video, a global average pool of $e$ on the final timestep, and a linear transformation of the resulting vector from 32 channels to 1 (see Table 5 for details). This CV-RNN model was trained to solve `FeatureTracker` by minimizing the following loss $\mathcal{L}_{total} = \mathcal{L}_{BCE} + \mathcal{L}_{synch}$ where:

$$\mathcal{L}_{BCE} = \text{BCELoss}(y, \hat{y}) \quad (3)$$

$$\mathcal{L}_{synch}(\theta) = \frac{1}{2}\left(\frac{1}{G}\sum_{l=1}^{G} V_l(\theta) + \frac{1}{2G}\left|\sum_{l=1}^{G} e^{i\langle\theta\rangle_l}\right|^2\right), \quad (4)$$

which involves minimizing (3) Binary Cross-Entropy between its predictions $\hat{y}$ and the label for each video $y$, and (4) maximizing the synchronization between groups of features (Ricci et al., 2021). To compute this second term, we first convolve the complex-hidden state $\phi_{xy}$ with real-valued weights that transform it from 32 channels to one, then compute the circular variance $V(\theta)$ and the average of $G$ phase groups $\langle\theta\rangle$. $V(\theta)$ is the intra-cluster synchrony, and minimizing it forces phase clusters to share the same value, whereas $e^{i\langle\theta\rangle}$ is the proximity between clusters, and minimizing it spreads the phase clusters on the unit circle (see SI A.8.1 for details). Overall, this loss induces neural synchrony in the complex-valued representation of the CV-RNN, ensuring that the target's features are bound with synchronized phases and separated from the distractors using desynchronized phases. We can also control the initialization of the complex-valued hidden state of attention $\phi[t]$ in the CV-RNN at the first timestep ($t = 0$), which affects its ability to learn to use synchrony (SI A.8.2 and Figs. A.5 and A.6 for ablation experiments on the complex operations). In all experiments described below, $\phi[0]$ is randomly initialized by sampling from a uniform distribution.

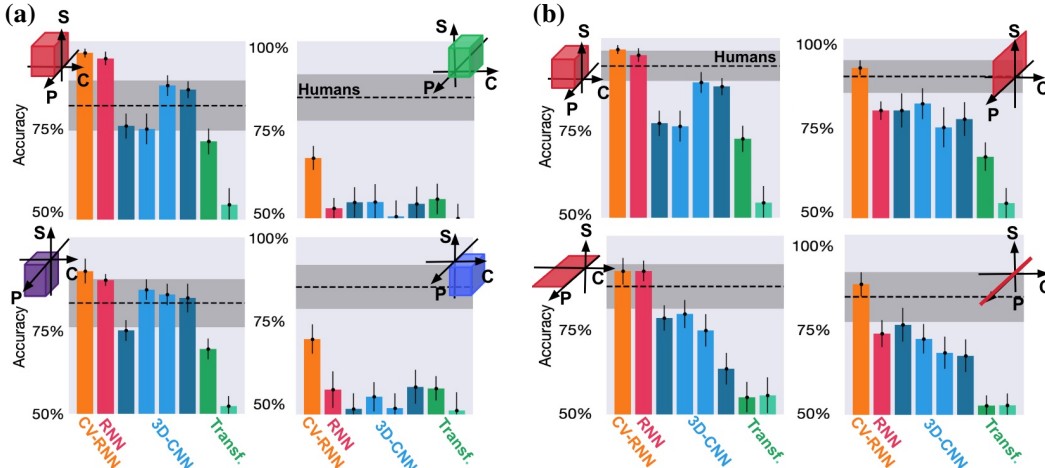

Figure 5: **Human and DNN performance on `FeatureTracker`**. (**a**) Humans and models are trained on videos where objects change in color and shape according to the distribution represented by the red cube. Both are then tested on videos where objects have appearances sampled from the same or different distributions. While humans are extremely accurate in each case, only the CV-RNN approaches their performance. (**b**) In a second experiment, we tested how humans and models perform on versions of the challenge where only the shape and position (top-right), color and position (bottom-left), or position alone (bottom-right; Linsley et al. (2021)) of objects change over time. Model performance and 95% confidence intervals, along with the mean (dotted line) and 95% confidence interval (grey box) of human performance are plotted for each condition. Darker bars indicate DNNs that were pre-trained, whereas lighter bars are DNNs trained from scratch. S=shape, P=position, C=color.

We compared the CV-RNN to humans and a sample of Visual Transformers and 3D Convolutional Neural Networks (3D CNNs) designed for video analysis. These models include the TimeSformer (Bertasius et al., 2021), MViT (Fan et al., 2021) (pre-trained on Kinetics400 (Kay et al., 2017)), ResNet3D (Tran et al., 2018), and MC3 (Tran et al., 2018). We included versions of the latter two 3D CNNs that were pre-trained on Kinetics400 and trained "from scratch." We also included the InT model from Linsley et al. (2021) in our analysis, which acted as a real-valued control for our CV-RNN (see SI A.9.1 and Tables 4, 5 for additional details and Fig.A.10 for a visualization of the computational efficiency of the CV-RNN).

All models were trained for 200 epochs on 100,000 videos sampled from the distribution described by the red cube in Fig. 4 (more details in SI A.9.3). Model performance was evaluated on a held-out set of 10,000 videos at the end of every epoch of training, and training was stopped early if accuracy on this set decreased for five straight epochs. We then took the weights of each model that performed best on this hold-out set and evaluated them on 10,000 videos from each condition depicted in Fig. 5.

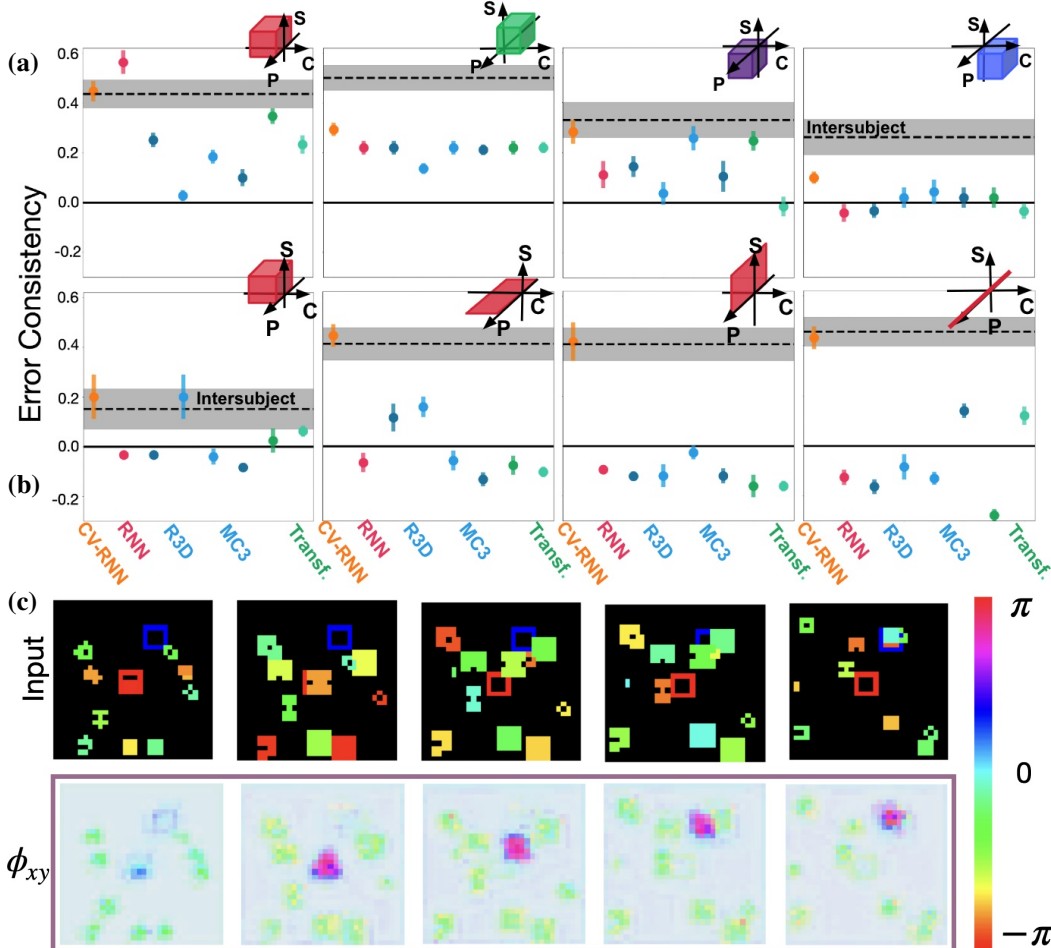

Figure 6: **Neural synchrony causes the CV-RNN to act more like humans than any other model tested.** Decision correlations of humans and models for each testing condition: features out-of-distribution conditions **(a)** and object evolution conditions **(b)**, computed using the Error consistency measure (Geirhos et al., 2020b). We present the same videos to humans and models and compare their errors. Each dot represents the error consistency averaged across human subjects. A higher number represents similar errors between humans and the model. The grey box represents the human inter-subject agreement. **(c)** Visualization of the phases of the hidden state of the complex attention mechanisms of our CV-RNN. The model affects a phase value for the target which is different from the one of the distractors and the background, and remains consistent across frames.

**CV-RNN accuracy significantly closer to humans than any other DNN tested.** All of the models that we tested, with the exception of the two based on visual Transformers, rivaled or exceeded human performance on videos with objects that were sampled from the same distribution as models and humans were trained on (Fig. 5a, top-left). The models performed similarly when tested on videos of objects with similar colors as training but different shapes (Fig. 5a, bottom-left). However, model performance fell precipitously when object colors were sampled from a different distribution at test time (Fig. 5a, right). In both of these conditions (same shapes/different colors, different shapes/different colors), the CV-RNN performed significantly better than the other models, which were close to chance accuracy. To further probe the abilities of models and humans to solve `FeatureTracker`, we generated versions where either the shape, color, or shape and color of objects were fixed to a single value. We first calibrated the performance of models and a new set of human participants against the first experiment by testing both on `FeatureTracker` videos where the color and shapes of objects varied according to the training distribution. In this case, humans were only

rivaled by the CV-RNN, InT RNN, and two of the 3D-CNNs (Fig. 5b, top-left). Only the CV-RNN's performance fell within the human confidence interval on the remaining conditions, which consisted of objects with fixed shapes but varying colors (Fig. 5b, bottom-left), varying shapes but fixed colors (top-right), or fixed colors and shapes (bottom-right; see SI Figs. A.12, A.13 and A.14 for extended benchmarks). We also include additional experiments with non-static background (Fig. A.17) and textured objects (Fig.A.18), as well as control experiments on the ability of the models to generalize to in-distribution features but out-of-distribution feature trajectories or to occlusions of the target in Figs. A.15 and A.16, as well as objects vanishing or moving in and out of the frame in Fig A.19.

**CV-RNN learns a human-like strategy to solve `FeatureTracker`.** To better understand how well our implementation of neural synchrony captures human strategies for object tracking, we next compared the errors it makes on `FeatureTracker` to humans. To test this, we turned to the method introduced by Geirhos et al. (2020a), which uses Cohen's κ to compute error correlations between observers while controlling for their task accuracy. For any given version of `FeatureTracker`, we then computed human-to-human (intersubject) error consistency as the average κ between all combinations of humans and model-to-human error consistency as the average κ between a model and all humans. CV-RNN errors were far more consistent with humans than other models (Fig. 6a). The CV-RNN's error consistency fell within the human intersubject interval on 6/8 versions of `FeatureTracker`, and was most similar to humans in the remaining two versions of `FeatureTracker` (expanded results in SI A.13.2, Figs.A.20 and A.21). The InT RNN was more consistent than the CV-RNN on `FeatureTracker` videos that were sampled from the same distribution as models were trained on (Fig. 6a, top-left), but was otherwise far less aligned with humans than the CV-RNN. In contrast, the remaining DNNs tended to make errors on different videos than humans. CV-RNN's ability to learn neural synchrony makes it more similar to human accuracy and decision-making on `FeatureTracker` than any other model we tested. To understand how the CV-RNN uses neural synchrony, we visualized the phase of the model's complex-valued hidden state $\phi_{xy}$ over `FeatureTracker` videos. The model learned to use the phase of its hidden state to track the position of targets independently of their color or shape. When the model correctly classified `FeatureTracker` videos, the phase of its attention smoothly tracked the target object and inhibited occluding distractors. When the model was incorrect, its phase tag jumped from object to object as it searched for the target (more examples in SI A.8.3).

## 5 DISCUSSION

Humans can track objects through the world even as they change in appearance, state, or visibility through occlusion. This ability underlies many everyday behaviors, from cooking to building, and as we have demonstrated, it remains an outstanding challenge for today's machine vision systems. However, by taking inspiration from neuroscience, we have developed CV-RNN, a novel recurrent network architecture that implements attention through neural synchrony, which can do significantly better. Our CV-RNN performs significantly better than any other DNN tested on `FeatureTracker`, and rivaled or approached human accuracy and decision-making on all versions of the challenge. Our findings are strongly related to neuroscience work, which suggests that phase synchronization is a key component of perceptual grouping. The *binding-by-synchrony* theory (Singer, 2007) suggests that when neurons synchronize their firing, the features they encode become bound to the same object. Thus, neural oscillations and the synchrony that they induce on neural populations may allow the brain to group features representing the same object in a visual scene (but see (Roelfsema, 2023; Shadlen & Movshon, 1999) for alternative hypotheses). Our work does not necessarily favor binding-by-synchrony over other competing theories (Fries, 2015; Jensen et al., 2014) but is a proof-of-concept that neural synchrony helps object tracking.

By visualizing the strategies learned by the CV-RNN to solve `FeatureTracker`, we were able to generate a novel and testable hypothesis regarding how neural synchrony supports object tracking. If neural synchrony acts similarly in human brains as it does in the CV-RNN, we speculate that similar phase behavior could be found in LFP recordings. Neuronal populations encoding for the target should display a phase shift on incorrect trials as we see within the CV-RNN. We also expect that neurons encoding for a distractor will be inhibited during occlusions, which could lead to a decrease in power of recorded LFPs (see Figs. A.9 and A.16 to observe this in the CV-RNN).

## 6 ACKNOWLEDGMENTS

Our work is supported by ONR (N00014-24-1-2026), NSF (IIS-2402875) to T.S. and ERC (ERC Advanced GLOW No. 101096017) to R.V., as well as "OSCI-DEEP" [Joint Collaborative Research in Computational NeuroScience (CRCNS) Agence Nationale Recherche-National Science Fondation (ANR-NSF) Grant to R.V. (ANR-19-NEUC-0004) and T.S. (IIS-1912280)], and the ANR-3IA Artificial and Natural Intelligence Toulouse Institute (ANR-19-PI3A-0004) to R.V. and T.S. Additional support was provided by the Carney Institute for Brain Science and the Center for Computation and Visualization (CCV). We acknowledge the Cloud TPU hardware resources that Google made available via the TensorFlow Research Cloud (TFRC) program as well as computing hardware supported by NIH Office of the Director grant S10OD025181.

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

# A    APPENDIX

## LIST OF FIGURES

## A.1    EXTENDED DISCUSSION

Our main discussion focuses on the link between the CV-RNN and Neuroscience theories. Here, we delve into additional technical aspects of our findings.

Firstly, we observed notably low accuracy in both Transformer models, even when the videos were sampled from a distribution similar to the training data. Despite the substantial advancements Vision Transformers have made for image classification (Dosovitskiy et al., 2021), they have minimal inductive biases for visual tasks, which may limit their effectiveness in small data video processing tasks (Tay et al., 2022). Indeed, the efficacy of self-attention is attributed to its capacity to densely route information within a context window, enabling the modeling of complex data. However, this

property entails fundamental drawbacks: an inability to model information beyond a finite window and quadratic scaling with respect to the window length. A current solution to this problem involves training these models on a very large amount of data. In this work, however, we maintain a fixed training set size across all models. Since all other architectural families performed well during training, and to avoid increasing the computational demands, we decided to limit the training set to $100,000$ samples.

Additionally, we observed that in several conditions, the models' performance exceeded human accuracy. This behavior might be attributed to the simple statistics of the data, which the models can easily learn. While humans can effortlessly translate their strategies to naturalistic videos, the scalability of the CV-RNN remains untested. We leave this investigation for future work with for example benchmarks proposed by Cai et al. (2023).

Finally, our benchmark includes state-of-the-art models but does not represent an exhaustive list of video classification models. We selected the models that we did to create a representative list of families and high-performing architectures in video processing. We release the code and dataset to enable the community to evaluate their own new models on the challenge and enhance the benchmark.

## A.2 LIMITATIONS

The CV-RNN performs similarly to humans on most testing sets, except for those where the color was unseen during training. We investigate the potential benefits of pretraining on the full colorspace in Fig. A.13 and consider extending this with self-supervised pre-training. Indeed, this pretraining improved the performance of the CV-RNN as well as the comparable RNN architecture, but the overall pattern of results remained the same.

Unlike the CV-RNN, humans are exposed to a wide variety of shapes and colors before learning the task. This prior knowledge is approximated by pre-training on Kinetics400 for some models, and a similar procedure could be beneficial for our circuit. The current model could also be improved by refining the initialization of the phases of $\phi[0]$ (see Figs.A.5,A.6) or conducting a hyper-parameter search to enhance optimization. Our primary goal is not to achieve the highest accuracy in every condition but to demonstrate our circuit as a robust proof of concept for neural synchrony through complex-valued units in tracking objects with changing appearances. This is validated by the similarity in performance and decision-making between humans and our model.

A final limitation of our work is that the CV-RNN is unable to learn to properly use phase for tracking without an additional loss function that enforces an effective phase synchronization strategy (see Figs.A.5,A.6). The development of complex-valued models where synchrony emerges as an unsupervised behavior is an active area of research (Löwe et al., 2022; Stanić et al., 2023). In this study, we examine the impact of synchrony on generalization abilities. We are optimistic that advancements in this research direction will lead to the development of unsupervised complex-valued models which, with a good strategy, will demonstrate human-like generalization abilities.

## A.3 BROADER IMPACTS

The primary goal of our study is to understand how biological brains function. `FeatureTracker` assists us in comparing models against human performance on a simple visual task, which tests visual tracking strategies when objects change appearance. The extension of the circuit with the implementation of neural synchrony allows us to make predictions about the type of neural mechanism that future neuroscience research might uncover in the brain. It is important to recognize that further development of this model has the potential for misuse, such as in surveillance. On the other hand, we believe our work is also beneficial for productive real-world applications such as self-driving cars and the development of robotic assistants. To promote research towards beneficial applications, we have open-sourced our code and data.

## A.4 COMPUTING

All the experiments of this paper have been performed using Quadro RTX 6000 GPUs with 16 Gb memory. The training time of each model is approximately 96 hours. We did not use extensive

hyper-parameter sweeps given the compute costs, but we did adjudicate between several approaches for inducing neural synchrony in our model.

## A.5 MOTIVATIONS

### A.5.1 SHELL GAME

Our motivating experiments for designing the CV-RNN were inspired by McLelland & VanRullen (2016), who computationally studied the role of oscillations and synchrony in different object segmentation strategies. We based our shell game off of their work. For this game, we generated two frames for each stimulus. For the first frame, we randomly selected two different colors and orientations from a pre-specified list, chose two non-overlapping sets of positions on a 24x24 pixel spatial map, and filled 4x4 pixel squares at those positions with colors and orientations to create objects. The second frame is generated in three different ways:

- Identical to the first frame, corresponding to class 0: no switch.
- One feature (either color or orientation) is randomly selected, and the positions of its values are swapped within its channel. This represents class 1: feature switch.
- The positions of the values in both channels are swapped simultaneously, representing class 2: object switch.

We generated a training set of $10,000$ samples with balanced classes. The task was a three-way classification problem, and all models were trained using gradient descent with Cross Entropy loss, the Adam optimizer (Kingma & Ba, 2014), and a learning rate of $3e-04$.

### A.5.2 2-LAYER ARCHITECTURES.

The model described in McLelland & VanRullen (2016) consists of a spiking network with two layers. The first layer simulates the primary visual cortex (V1), capturing low-level retinotopic information. The second layer, which represents the inferotemporal cortex (IT), has a global receptive field, where all cells receive input from the entire spatial area of the first layer. We constrained our model with this overarching structure to identify challenges that biological visual systems face when tracking objects that change appearances over time.

Stimuli in our shell game can take four orientations and three colors, resulting in 12 possible feature combinations. Consequently, the second layer contains 12 neurons. Due to this architectural design, the model is incapable of binding colors and orientations at their respective positions, which is many more combinations than the model is capable of representing.

We constructed a simple two-layer network tailored for our task. The architecture, illustrated in Table 1 (also see number of parameters and flops in Table 2), includes an initial convolutional layer that transforms the input into a spatial map of the stimuli, analogous to the first layer of the spiking model described earlier. To remove spatial information and create a second, high-level layer, we apply a MaxPooling operation. The second layer, which can be either a linear or RNN layer, resembles the high-level layer of the spiking network described in McLelland & VanRullen (2016). It is also constrained to have a number of neurons equal to the number of possible feature conjunctions. We also experimented with another architecture where the MaxPooling operation was omitted, allowing the second layer to receive inputs from all neurons in the first layer. The results from this architecture were identical to those obtained with the initial design.

Given that our input consisted of two frames, we passed each frame separately through the network and stored the representation from the linear layer, which ostensibly represents feature conjunctions. To make a prediction about the type of switch observed between the two frames, we introduced a classification readout layer. Similar to the findings of McLelland & VanRullen (2016), these models were not able to disambiguate the three different classes (see Fig. A.1); they were not able to distinguish the feature class from the object switch class.

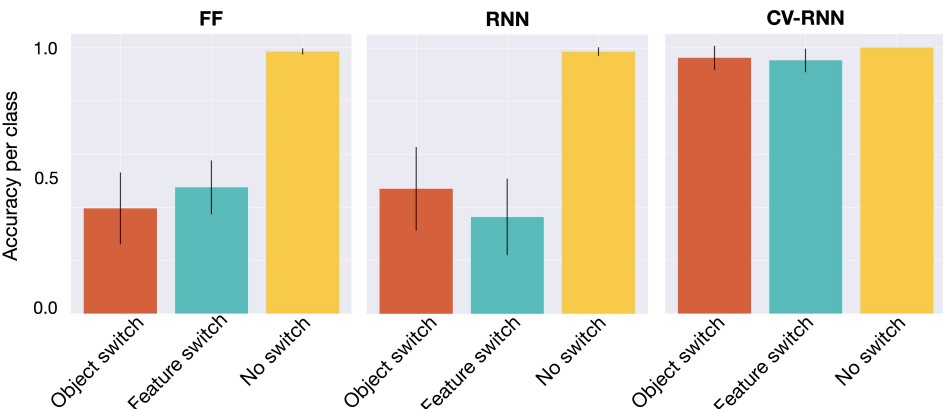

Figure A.1: Accuracy per class for each of the tested models (Feedforward network, RNN, CV-RNN), on the task defined in Sec. A.5.1. Each model was trained with 5 different initializations. We report the mean performance associated with the standard error of the models on a separate test set. The orange bar represents the accuracy on images where both features (color and orientation) switched position simultaneously. The turquoise bar shows the performance of the models on images where only one of the two attributes switched positions. Finally, the yellow bar stands for images where both frames are identical.

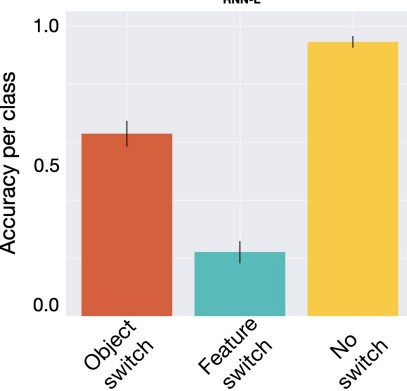

Figure A.2: Performance of a large RNN matching the number of parameters of the CV-RNN on the shell game.

| Layer | Input Shape | Output Shape |
|---|---|---|
| *Conv2d* | [2,24,24] | [12,24,24] |
| *MaxPooling* | [12,24,24] | [12,1,1] |
| **Linear/RNN** | [12,1,1] | [12] |
| Linear | [12*2] | [3] |

Table 1: Architecture of the feed-forward network for the `shell game`. The italic layers reproduce the architecture of the spiking network from (McLelland & VanRullen, 2016). The bold layer represents the two studied architectures (FF network or RNN). The last Linear layer receives inputs from the second layer after processing the two frames separately.

| Model | #Params | Flops |
|---|---|---|
| CV-RNN | 1,269 | 34,943,616 |
| RNN | 835 | 10,695,680 |
| RNN-L | 6451 | 35,240,192 |

Table 2: Number of parameters and flops of each model used in the shell game.

## A.6 CV-RNN

### A.6.1

The InT circuit. Our CV-RNN was derived from the InT circuit (Linsley et al., 2021), which was inspired by neuroscience and designed for tracking objects based on their motion. Our goal was to enhance this circuit to become tolerant to changes in object features over time.

The full circuit represents two interacting inhibitory and excitatory units defined by (but see Linsley et al. (2021) for more details):

$$i[t] = gi[t-1] + (1-g)[z[t] - (\gamma i[t]a[t] + \beta)m[t] - i[t-1]]_+ \tag{5}$$

$$e[t] = he[t-1] + (1-h)[i[t] + (\nu i[t] + \mu)n[t] - e[t-1]]_+ \tag{6}$$

and a mechanism for selective "attention":

$$a[t] = \sigma(\mathbf{W_a} * e[t-1] + \mathbf{W_z} * z[t]) \tag{7}$$

where

$$m[t] = \mathbf{W_{e,i}} * (e[t-1] \odot a[t]) \quad \text{and} \quad n[t] = \mathbf{W_{i,e}} * i[t] \tag{8}$$

and

$$g[t] = \sigma(\mathbf{W_g} * i[t-1] + \mathbf{U_g} * z[t]) \quad \text{and} \quad h[t] = \sigma(\mathbf{W_h} * e[t-1] + \mathbf{U_h} * i[t]) \tag{9}$$

Here, $z[t]$ denotes the input at frame $t$, which is subsequently forwarded to the inhibitory unit $i$ interacting with the excitatory unit $e$. Both units possess persistent states preserving memories facilitated by gates $g$ and $h$. Additionally, the inhibitory unit is modulated by another inhibitory unit, $a$, which operates as a non-linear function of $e$ capable of modulating the inhibitory drive either downwards or upwards (i.e., through disinhibition). In essence, the sigmoidal nonlinearity of $a$ enables position-selective modulation, which we refer to as "attention". Furthermore, as $a$ is contingent on $e$, lagging temporally behind $z[t]$, its activity reflects the displacement (or motion) of an object in $z[t]$ versus the current memory of $e$. Fundamentally, this attention mechanism aims to relocate and enhance the target object in each successive frame.

### A.6.2 COMPLEX OPERATIONS.

Before delving into our methodology for developing the CV-RNN, we first examined various operations achievable with complex numbers, including weights and operations. Given the vast array of potential operations, we will only elaborate on those that will be utilized throughout the remainder of this article.

Considering a complex number $z \in \mathbb{C}$, $z$ can be written as:

$$z = Real(z) + j.Imag(z) \quad \text{or} \quad z = ||z||.e^{j.\theta_z} \tag{10}$$

Where *Real* and *Imag* respectively denote the real and imaginary parts of the complex number, with $j^2 = -1$. Similarly, $||.||$ and $\theta$ respectively stand for the magnitude and the phase of the complex number.

Applying complex weights on real-valued activation. Given a real-valued activation $x$ and a set of complex weights $\mathbf{W}^C$, the resulting activity $z_1$ is:

$$z_1 = \mathbf{W}^C * x = Real(\mathbf{W}^C) * x + j.Imag(\mathbf{W}^C) * x = (||\mathbf{W}^C|| * x).e^{j.\theta_{\mathbf{W}^C}} \tag{11}$$

The intuition behind the use of this operation is to learn an appropriate phase distribution ($\theta_{\mathbf{W}^C}$) from the real-valued input.

Applying real-valued weights on complex activation. Given a complex-valued activation $z_{in}$ and a set of real weights $\mathbf{W}$, the resulting activity $z_2$ is:

$$z_2 = \mathbf{W} * z_{in} = Real(z_{in}) * \mathbf{W} + j.Imag(z_{in}) * \mathbf{W} = (||z_{in}|| * \mathbf{W}).e^{j.\theta_{z_{in}}} \tag{12}$$

Contrary to Eq. 11, the assumption here is that the phase of $z_{in}$ remains unchanged, but the amplitude is updated by the weights.

### A.6.3   COMPLEX ATTENTION MECHANISM.

To induce neural synchrony through complex-valued units, in the attention mechanism of the RNN, we began from Eq. 7 and proceeded in three steps:

1. **Convert $e$ and $z \in \mathbb{R}$ to complex numbers:** we apply complex weights $\mathbf{W_a^C}$ and $\mathbf{W_z^C}$ to $e$ and $z$ respectively following the operation described in Eq. 11. We obtain $e_c$ and $z_c \in \mathbb{C}$.

2. **Update $\phi$ with the current feed-forward drive:** we apply real weights $\mathbf{W_a}$ to $(z_c + \phi)$ with Eq. 12. The addition operation activates the pixels corresponding to the new position of the target within the complex hidden state. Subsequently, applying the real weights deactivates the pixels where the target is no longer present. The initialization of $\phi$ at $t = 0$ is detailed below.

3. **Compute the new complex attention map:** by combining the information for the feed-forward drive, the excitation unit, and the complex hidden state $a = z + e + \phi$.

4. **Transfering the activity back to the real domain and applying the sigmoid operation:** because, by definition, the amplitude of a complex number is positive and we want to apply a sigmoid on the attention map, we first normalize $a$ using the InstanceNorm (*In*) operator (Ulyanov et al., 2016). The final attention map is obtained with $a = \sigma(In(a))$.

5. **Obtaining a 2D phase-map of $\phi$ (2D case only).** We additionally apply a real convolution (Eq. 12) on $\phi$ to reduce the channel dimension and obtain a phase map of the complex hidden state: $\theta = arg(\mathbf{W_p} * \phi)$, where $arg(.)$ stands for the phase of the complex number.

### A.6.4   SOLVING THE SHELL GAME WITH THE CV-RNN.

We embedded our CV-RNN into the hierarchical model of a biological visual system described in Table 3. We introduced the phase information from the first layer to ensure that the phases can bind position and features. The *ComplexConv2d* uses the operator defined in Eq. 12. The complex-valued input combined the amplitude and a phase map initialized randomly for each frame. The *ComplexMaxPooling* operation applies a standard MaxPooling on the amplitude of the complex activation and retrieves the phases associated with the amplitude propagated to the next layer.

| Layer | Input Shape | Output Shape |
|---|---|---|
| *ComplexConv2d* | [2,24,24] | [12,24,24] |
| *ComplexMaxPooling* | [12,24,24] | [12,1,1] |
| **CV-RNN** | [12,1,1] | [12] |
| Linear | [12*2] | [3] |

Table 3: The architecture of the complex-valued model adapted from Tab. 1 to introduce phase information into the model. The operations are now all complex except for the classification layer, receiving input from the output of the CV-RNN converted to the real-valued domain in the attention mechanism.

### A.7 FEATURETRACKER

#### A.7.1 GENERATING OBJECT TRAJECTORIES.

To generate objects with smoothly changing appearances, we employed three distinct rules for generating trajectories within each dimension of the feature space:

- **Position**: Following the approach in Linsley et al. (2021), spatial trajectories are randomly generated, commencing from a random position within the first input frame. To maintain trajectory smoothness, an angle is randomly selected. If this angle falls within a predefined range ensuring trajectory smoothness, the object advances in that direction; otherwise, it remains stationary.

- **Color**: Colors are generated using the HSV colorspace, with Saturation and Value fixed. Initialization begins with a random Hue value. Subsequently, at each frame, the Hue is updated at a constant speed and in a consistent direction across all objects.

- **Shape**: Objects are represented by 5x5 squares. They begin in a random state, where a random number of pixels within this grid are active. Over time, they evolve according to the rules of the Game of Life (GoL) (Gardner, 1970):

  - If the pixel is active (value 1), and the number of active neighbors is less than 2 or more than 3: the pixel becomes inactive (value 0).
  - If the pixel is inactive, and the number of active neighbors is equal to 3: the pixel gets active.
  - We add a third rule to avoid making the objects disappear: if no pixel is active, the center pixel is activated.

#### A.7.2 GENERATING SEVERAL CONDITIONS TO EVALUATE GENERALIZATION.

We divided the challenge into 10 different conditions. The training condition was generated with (i) half the HSV spectrum (and a fixed Saturation and Value) for the colors, (ii) the first (and last) rule of the GoL to generate the shapes – making the objects grow over time (see Fig. A.3 – top row for illustrations).

Next, we introduced testing conditions where features are out-of-distribution (OOD), meaning colors and/or shapes were not encountered during training. These conditions are depicted in the second row of Fig. A.3, and are as follows:

- **OOD colors:** Shapes and positions are sampled identically to the training distribution. However, colors are drawn from the unobserved portion of the colorspace.

- **OOD shapes:** Colors and positions evolve in a manner similar to the training distributions. Shapes, however, evolve according to the second and last rules of the Game of Life (GoL), resulting in their sizes diminishing over the duration of the video.

- **OOD colors and shapes:** Both the shapes and the colors are out-of-distribution (following the two rules described above). The position is the sole common feature retained from the training videos.

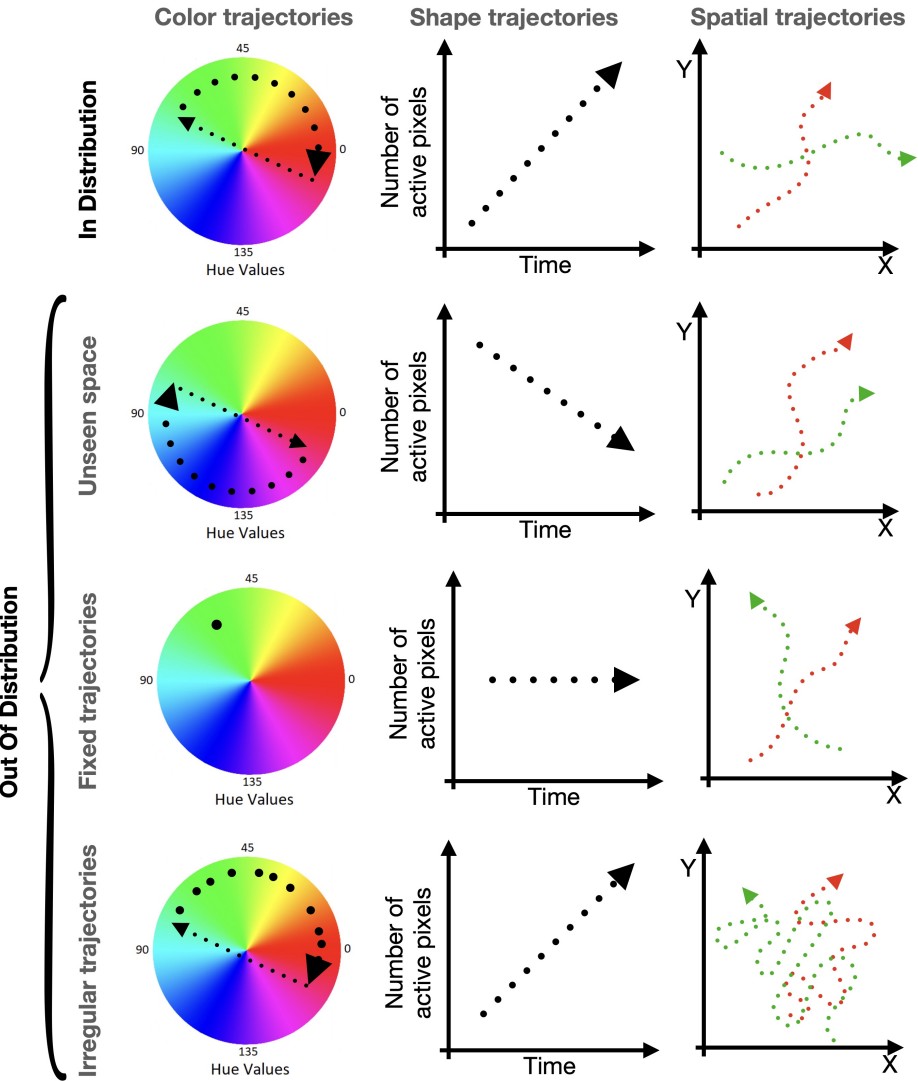

Figure A.3: The trajectories in each condition are devised to assess the models' generalization capabilities. The in-distribution trajectories, depicted in the first row, exhibit smooth color sampling within half of the HSV spectrum. Shapes evolve based on the first rule of the Game of Life (Gardner, 1970), while positions are generated similarly across the main conditions. Out-of-distribution conditions introduce variations either in the feature sampling space, the temporal evolution of objects, or the speed of change over time.

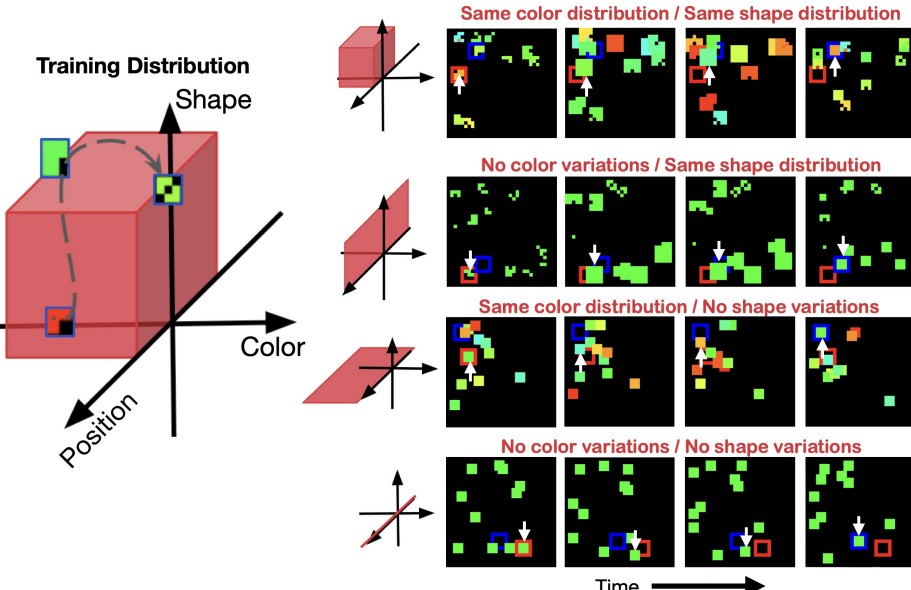

Figure A.4: `FeatureTracker` also encompasses conditions where the trajectory in either dimension (shape or color) lies within the training distribution but remains constant across frames. In the second row, the depicted condition features objects with a fixed green coloration. Below, objects transition between colors while maintaining fixed shapes (squares). Lastly, the last condition mirrors the PathTracker challenge (Linsley et al., 2021), wherein all objects are green squares.

Thirdly, videos are generated where the trajectory of shapes and/or colors lies within the training feature space but remains fixed over time (refer to Fig.A.3, third row, and Fig.A.4 for visual representations):

- **Fixed trajectory colors:** All objects maintain a consistent green color throughout the video, with no changes over time. Meanwhile, their shapes evolve according to the rules utilized to generate the training videos.

- **Fixed shape trajectory:** All objects are defined by 3x3 squares, and their shapes remain constant throughout the video without any alterations over time. Meanwhile, their colors are generated in a manner similar to the training data.

- **Fixed colors and shapes:** All the objects are 3x3 green squares (akin to the PathTracker challenge (Linsley et al., 2021)).

In the last six out-of-distribution conditions (features or trajectories), the position (spatial trajectory) is the only feature sampled identically to the training distribution.

We additionally generated a final testing set comprising two conditions with videos of objects whose colors and/or spatial trajectories are irregular (in contrast to the smooth and predictable patterns observed during training), as illustrated in Fig. A.3, last row:

- **Irregular colors:** Within the same colorspace as the training data, we randomize the speed of change of the Hue. Consequently, the colors become an unreliable feature for tracking the target.

- **Irregular positions:** The range of permissible angles for each step of the object's spatial trajectory is expanded compared to the training phase. As a result, the spatial trajectories become more erratic, making it more challenging to track the target.

- **Irregular colors and positions:** This testing set combines the two conditions described above, resulting in neither the colors nor the positions being as predictable as they were during training.

## A.8 CV-RNN AND FEATURETRACKER

### A.8.1 SUPERVISING THE PHASE SYNCHRONY.

To induce phase synchrony in the CV-RNN, we added a synchrony loss applied on the phases ($\theta_{xy}$ in Fig. 3). Eq. 4 is derived from the general case initially proposed by Sepulchre et al. (2008) and used to synchronize a population of oscillators by Ricci et al. (2021):

$$\mathcal{L}(\theta) = \frac{1}{2}\left(\frac{1}{k}\sum_{l=1}^{k} V_l(\theta) + S(\theta)\right) \tag{13}$$

where $V_l$ is the circular variance of the $l^{\text{th}}$ group and S is a loss term aimed at promoting splayness (Strogatz & Mirollo, 1993) among the target groups. This ensures that the mean phases within groups are evenly distributed around the unit circle. Let $\langle\theta\rangle_l$ denote the average phase of an oscillatory population, then we set:

$$S = \sum_{g=1}^{\lfloor k/2 \rfloor} \frac{1}{2g^2 k}\left|\sum_{l=1}^{k} e^{ig\langle\theta\rangle_l}\right|^2 \tag{14}$$

Eq. 14 guarantees that the centroids of the phase groups are equidistant on the unit circle. Coupled with $V_l$ in Eq. 13, ensures uniformity of phases within the groups, this loss induces "synchrony" in the phase population $\theta$. During the generation of input videos, we also created a segmentation mask for each frame, which distinguished the target object from the distractors, as well as from the start and end markers, and the background. We considered 3 groups ($G = 3$): target, distractors, and background. The group containing the start/end markers was excluded from the loss calculation; hence, the model was unrestricted in its placement of these markers within any of the explicitly defined groups. This loss was applied at each frame, with the loss values accumulated over time. The sum over time is then combined with the Binary Cross-Entropy (BCE) Loss for classification. The general loss is:

$$\mathcal{L}_{\text{CV-RNN}} = \text{BCELoss}(y, \hat{y}) + \sum_{t=0}^{T} \mathcal{L}_{synch}(\theta_{xy}[t]) \tag{15}$$

with $T = 32$, $y$ the ground truth and $\hat{y}$ the prediction of the model. We do not enforce phase consistency between frames. However, the model independently learned to maintain similar phase values for each group across frames (see Figs 6c), A.7 and A.8).

### A.8.2 INITIALIZATION OF THE COMPLEX-HIDDEN STATE $\phi_{xy}$

Looking at Eq. 2, one might wonder how to initialize $\phi[t]$ at $t = 0$. We tested a variety of different strategies.

We can use the aforementioned masks to initialize the phases of the hidden state. This initialization involves generating four phase values, which are equidistant on the unit circle. Each phase value is assigned to a different group in the mask (target, distractors, start/end marker, background). This initialization method is referred to as "Phase Segmentation/First Frame" in Figs. A.5 and A.6, and it represents the "best" solution induced in the model at the first timestep. The challenge lies in maintaining this solution over time. We also experimented with learning this phase initialization by using a convolution on the input frame at $t = 0$ to initialize the complex hidden state, referred to as "Learnable Phases/First Frame" in the Figures. The model presented in the main results is initialized with random phases ("Random phases/First Frame") providing a fairer comparison with the baselines that use less information and fewer parameters than the two models described previously. We also include two negative controls: one model trained without the synchrony loss (see paragraph above), and another model where the phases are randomized at each timestep. The first model represents a complex-valued model employing a free strategy, which may not be well-suited to the task. The second model lacks recurrent phase information and, therefore, cannot maintain phase-based tracking of the target.

As expected, the model with phase segmentation initialization consistently outperformed others in each condition. The model with learnable initialization also performed well across most conditions, except for OOD colors. Surprisingly, the model with random phases demonstrated remarkable generalization abilities and achieved performance close to models with more information or parameters.

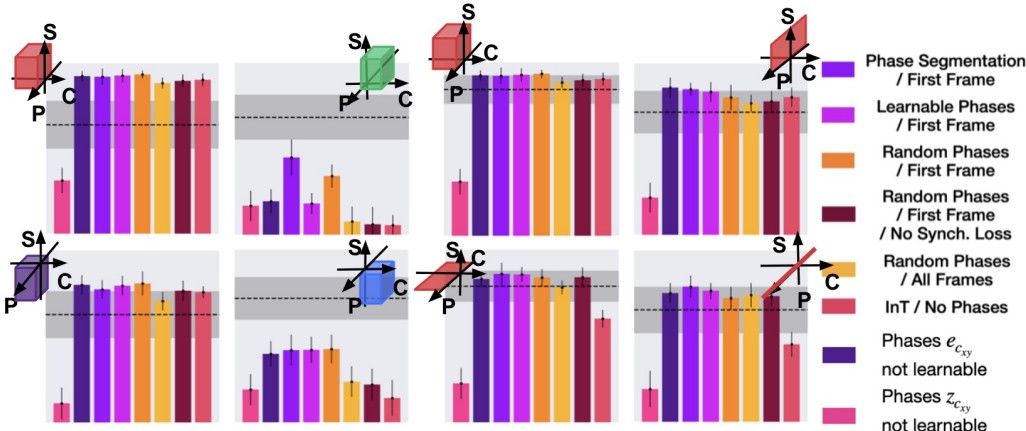

Figure A.5: The phase initialization at the first or at each timestep can be manipulated and affect the generalization abilities. Each bar represents a different phase strategy and its impact on the test performance on the conditions with features out of the training distribution – akin to a systematic ablation study of the phase strategy in the CV-RNN.

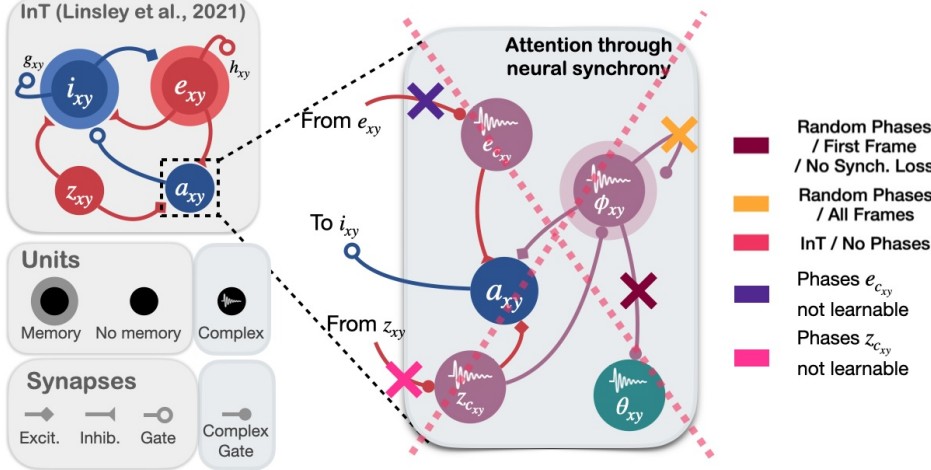

Figure A.6: Illustration of the location of the ablations performed in Fig. A.5 in the CV-RNN.

These results suggest that there is potential for further improvement in bridging the gap with human performance by providing informative phase initialization to the models. Finally, both negative controls confirm the necessity of an explicit objective and consistent phase information to consistently achieve human-level performance.

### A.8.3 VISUALIZING THE PHASE STRATEGY.

In Figs. A.7 and A.8, we show visualization of the phases of $\phi_{xy}$ and $\theta_{xy}$ for each condition. We pick random videos for each test set and show frames equally sampled between the first and the last. The spatial map illustrating $\phi_{xy}$ is derived by taking a complex average across the channel dimension. Additionally, we utilize the amplitude as the alpha value, thereby masking out the phases representing the background and highlighting the objects in each frame. The spatial map $\theta_{xy}$ is the unit on which the synchrony loss is applied (Eq. 4). For this reason, it distinctly demonstrates a detailed separation between groups. However, the model still struggles to identify the target in conditions where the color is out-of-distribution (refer to Fig. A.7, where the green cube and blue cube are depicted). In such cases, the phase value corresponding to the target (red) jumps position between frames, as if the

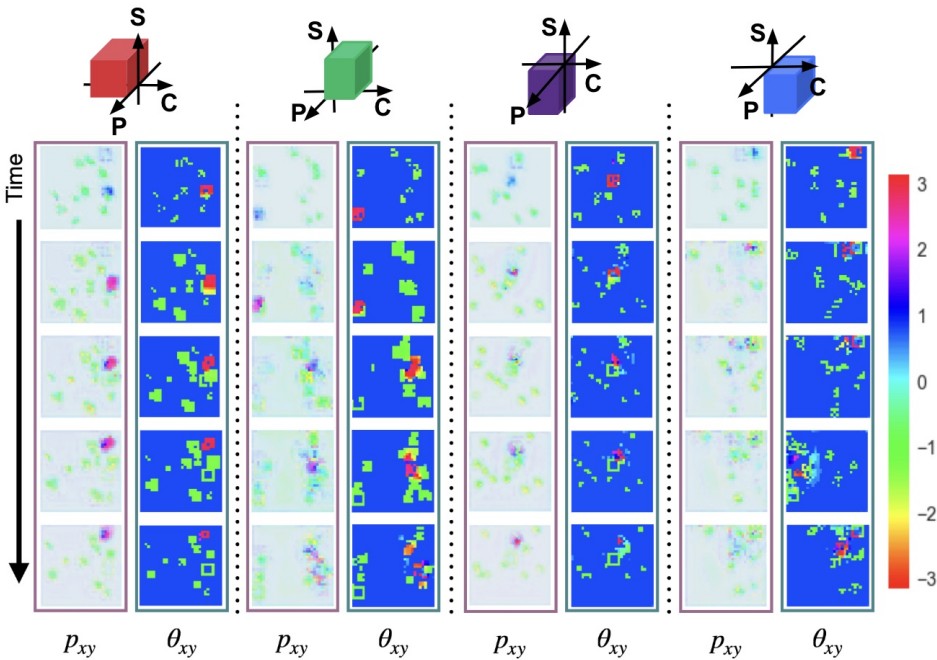

Figure A.7: Visualizations of $\phi_{xy}$ average across channels and masked out by the complex amplitude, and $\theta_{xy}$ on which the synchrony loss is applied, on conditions where the features are out-of-distribution. In conditions where the color is out-of-distribution, the model struggles to keep track of the target. However, when the shapes were unseen during training, the model can behave similarly than during training.

model were searching for the target among the variety of objects in the frame. Nonetheless, in the other instances, the target is clearly discerned from the distractors, even when it is occluded by them.

## A.9 MODELS

### A.9.1 BASELINE MODELS FOR OUR BENCHMARKING.

We choose representative models of the tracking literature from each big family of vision models:

- **3D-CNNs:** ResNet18-based type of model for video that employs 3D convolutions (Tran et al., 2018). We utilize two versions of the model: the standard R3D and MC3, a variant that employs 3D convolutions only in the early layers of the network while employing 2D convolutions in the top layers. Both versions of these models are trained from scratch or pre-trained on Kinetics400 (Kay et al., 2017).

- **Transformers:** We employ the latest state-of-the-art spatio-temporal transformer, MViT (Fan et al., 2021). MViT is a transformer architecture designed for modeling visual data such as images and videos. Unlike conventional transformers, which maintain a constant channel capacity and resolution throughout the network, Multiscale Transformers feature multiple channel-resolution scale stages. We experimented with a version of the model trained from scratch, but it failed to learn the task. Therefore, we only report results for the pre-trained version on Kinetics400. Additionally, we include another state-of-the-art transformer architecture: TimeSformer (Bertasius et al., 2021). TimeSformer is a convolution-free approach to video classification, relying solely on self-attention over space and time. It adapts the standard Transformer architecture to videos by facilitating spatiotemporal feature learning directly from a sequence of frame-level patches. We exclusively utilize a version of the model trained from scratch.

- **RNN:** We include the latest state-of-the-art RNN for tracking: InT (Linsley et al., 2021) (see description of the model in Section A.6.1).

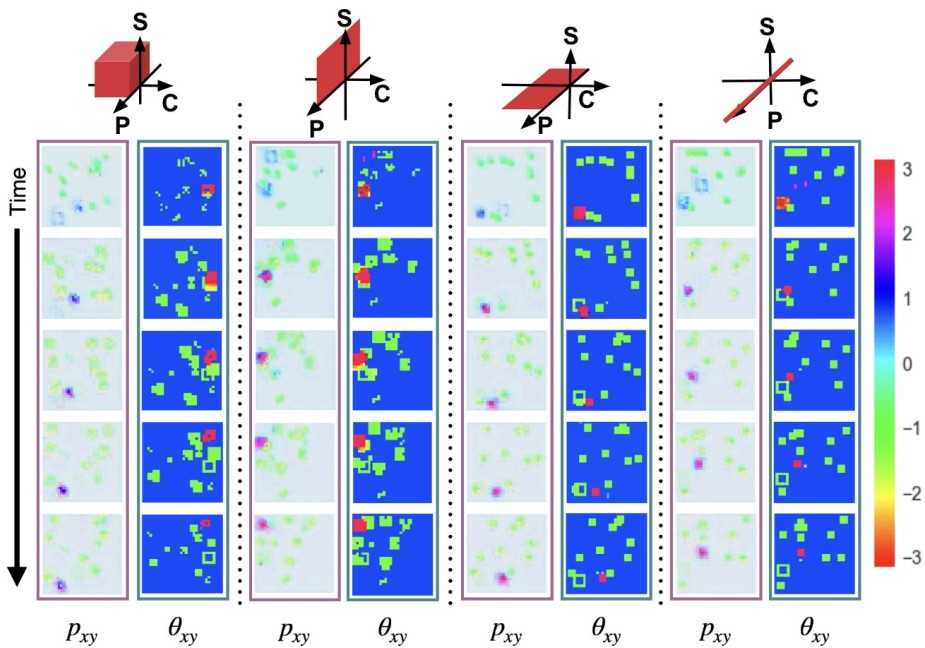

Figure A.8: Visualizations of $\phi_{xy}$ average across channels and masked out by the complex amplitude, and $\theta_{xy}$ on which the synchrony loss is applied, on conditions where the trajectory of features over the course of the video are fixed. Even though the dynamic of the objects across time was unseen, the model is able to adopt the same strategy as during training and keep track of the target.

### A.9.2 EMBEDDING THE RNN AND CV-RNN INTO A BINARY CLASSIFICATION ARCHITECTURE.

The RNN and our CV-RNN are circuits integrated into a larger architecture to preprocess the input frames and generate classification predictions. This architecture is detailed in Table 4.

| Layer | Input Shape | Output Shape |
|---|---|---|
| Conv3D | [3,32,32,32] | [32,32,32,32] |
| $\forall t \in \{0,...,31\}$: RNN/CV-RNN | [32,1,32,32] | [32,1,32,32] |
| Conv2d | [32,32,32] | [1,32,32] |
| Conv2d | [2,32,32] | [1,32,32] |
| AvgPool2d | [1,32,32] | [1,1,1] |
| Linear | [1] | [1] |

Table 4: Full architecture including the RNN/CV-RNN circuits. The input video is pre-processed by a 3D convolution. Each frame is passed one after the other into the circuits. The excitation state of the last frame is passed to a readout 2D convolution. This output is concatenated with the input and processed by another convolution charged to assess whether the target is inside the end marker. The spatial information is reduced by an AveragePooling2d before getting the prediction of the model via a Linear layer.

The number of parameters for each architecture in our benchmark is summarized in Table 5. The RNN and CV-RNN employ significantly fewer parameters than the other architectures in our benchmark. The CV-RNN, with its additional operations in the attention employing neural synchrony, contains slightly more parameters than the RNN. To ensure a fair comparison, we conduct a control experiment by increasing the number of parameters in the RNN to match that of the CV-RNN. We demonstrate that the results remain unchanged in this scenario (refer to Fig. A.12).

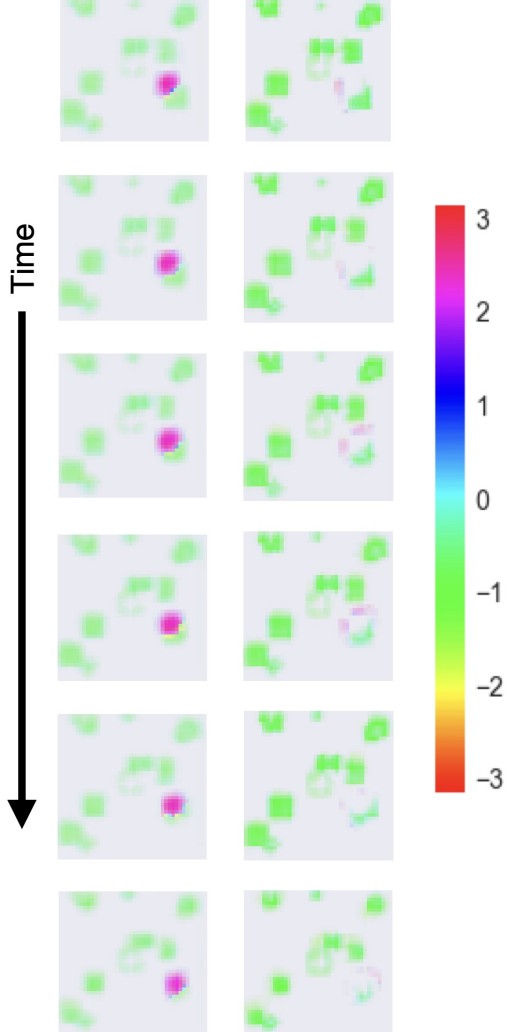

Figure A.9: Visualizations of two distinct channels of $\phi_{xy}$ and masked out by the complex amplitude. One of the channels encodes the target and the distractors while the second encodes only for the distractor. During an occlusion, the power of the neurons encoding the distractor occluding the target is shut down to privilege the target.

| Model | #Params |
|---|---|
| CV-RNN | 171,580 |
| RNN | 108,214 |
| RNN-L | 177,010 |
| R3D | 33,166,785 |
| MC3 | 11,490,753 |
| MViT | 36,009,697 |
| TimeSformer | 189,665 |

Table 5: Number of parameters of each model used in our benchmark.

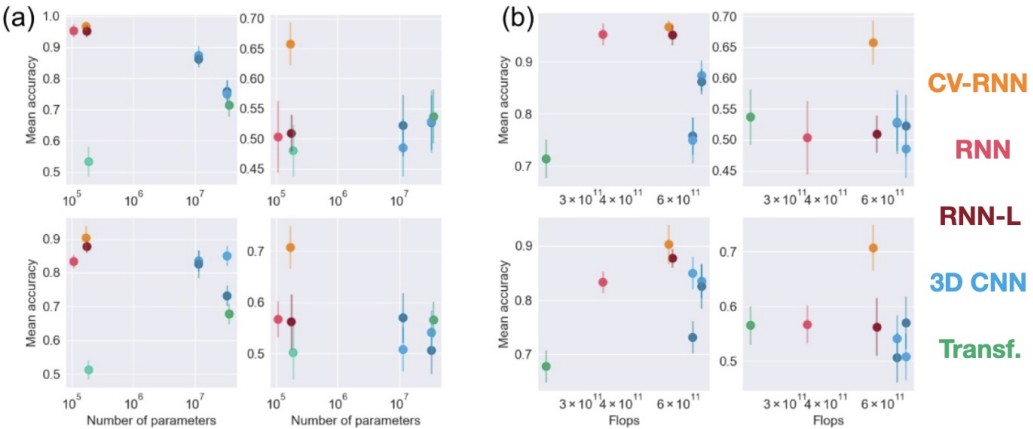

Figure A.10: **Computational efficiency of the models and test on occlusions.** Accuracy vs. (a) number of parameters and (b) flops on the different conditions of `FeatureTracker`.

### A.9.3 TRAINING DETAILS.

We use an identical training pipeline for all the models. This pipeline includes a training set composed of $100,000$ videos of 32 frames and 32x32 spatial resolution. We employ the Adam optimizer (Kingma & Ba, 2014) with a learning rate of $3e - 04$, a batch size of 64 during 200 epochs with a Binary Cross-Entropy loss.

### A.10 HUMAN BENCHMARK

For our benchmark experiments, we recruited 50 participants via Prolific, each of whom received $5 upon successfully completing all test trials. Participants confirmed completion by pasting a unique system-generated code into their Prolific accounts. The compensation amount was determined by prorating the minimum wage. Additionally, we incurred a 30% overhead fee per participant paid to Prolific. In total, we spent $325 on these benchmark experiments.

### A.10.1 EXPERIMENT DESIGN

At the beginning of the experiment, we obtained participant consent using a form approved by a university's Institutional Review Board (IRB). The experiment was conducted on a computer using the Chrome browser. After obtaining consent, we provided a demonstration with clear instructions and an example video. Participants also had the option to revisit the instructions at any time during the experiment by clicking on a link in the top right corner of the navigation bar.

Participants were asked to classify the video as "positive" (the target leaving the red marker entered the blue marker) or "negative" (the target leaving the red marker did not enter the blue marker) using the right and left arrow keys respectively. The choice for keys and their corresponding instances were mentioned below the video on every screen (See Fig. A.11). Participants were given feedback on their response (correct/incorrect) after every practice trial, but not after the test trials.

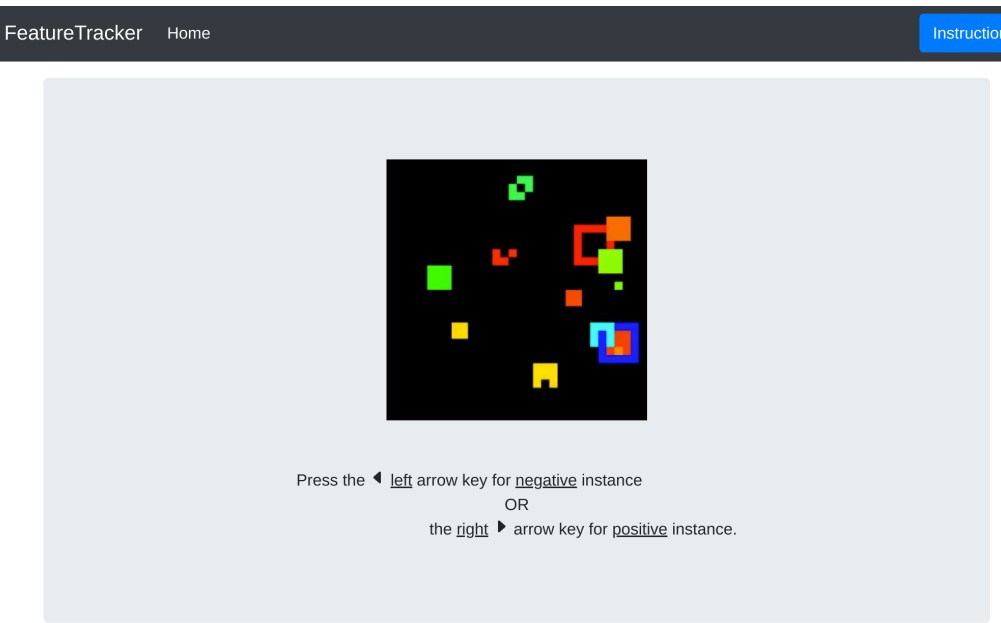

Figure A.11: An experimental trial screen.

The experiment was not time-bound, allowing participants to complete it at their own pace, typically taking around 20 minutes. Videos were played at 10 frames per second. After each trial, participants were redirected to a screen confirming the successful submission of their responses. They could start the next trial by clicking the "Continue" button or pressing the spacebar. If they did not take any action, they were automatically redirected to the next trial after 1000 milliseconds. Additionally, participants were shown a "rest screen" with a progress bar after every 40 trials, where they could take additional and longer breaks if needed. The timer was turned off during the rest screen.

### A.10.2 STATISTICAL TESTS

We performed statistical tests on human accuracy to validate that the subjects were performing significantly above chance. For each testing condition, we perform a binomial test considering the total number of trials, the number of successes (sum across subjects), and a chance level of 50%:

- Colors/Shapes from the same distribution: with 511 trial and 417:
  $p = 1.4971940604135627e - 49$,
  *Hit rate* $= 0.8235294117647058$,
  *False alarm rate* $= 0.19140625$,
  *D-prime* $= 1.8016253862743108$,
  *D-prime shuffled* $= -0.20203417784264527$ with *Standard deviation* $= 0.22711300145337426$,
  *Mean RT across all subjects* $= 8.930597560048328$ with *Standard deviation* $= 0.7095593260185693$.

- Colors from a different distribution but shapes from the same distribution: with 510 trial and 426:
  $p = 4.372661934686906e - 56$,
  *Hit rate* $= 0.9450980392156862$,
  *False alarm rate* $= 0.27450980392156865$,
  *D-prime* $= 2.1983049458366786$,
  *D-prime shuffled* $= 0.09232866338436123$ with *Standard deviation* $= 0.1609848258114674$,
  *Mean RT across all subjects* $= 8.595721796447155$ with *Standard deviation* $= 0.35099136986358925$.

- Colors from the same distribution but shapes from a different distribution: with 518 trial and 420:

$p = 1.780269626788243e - 48$,
*Hit rate* $= 0.84375$,
*False alarm rate* $= 0.22137404580152673$,
*D-prime* $= 1.7775510822035647$,
*D-prime shuffled* $= -0.1375093584656358$ with *Standard deviation* $=$ 0.17728478748933665,
*Mean RT across all subjects* $= 9.304951464300643$ with *Standard deviation* $=$ 0.6846367945164648.

- Colors and shapes from a different distribution: with 515 trial and 441:
$p = 1.286820789401082e - 64$,
*Hit rate* $= 0.867704280155642$,
*False alarm rate* $= 0.15503875968992248$,
*D-prime* $= 2.130663946673155$,
*D-prime shuffled* $= -0.15882686267724502$ with *Standard deviation* $=$ 0.1795164617241738,
*Mean RT across all subjects* $= 9.271893953567382$ with *Standard deviation* $=$ 0.7901287887031236.

We proceed similarly for the additional conditions resulting in the following p-values:

- Colors/Shapes evolving similarly: with 400 trial and 368, $p = 1.6626425479283706e - 73$.

- Colors fixed to green and shapes evolving similarly: with 401 trial and 358, $p = 6.064324617765989e - 63$.

- Colors evolving similarly to the training distribution but shapes fixed to 3x3 squares: with 401 trial and 343, $p = 2.4917696306121515e - 50$.

- Colors and shapes fixed to 3x3 green squares: with 401 trial and 317, $p = 6.243000914755226e - 33$.

### A.11 CONTROL EXPERIMENTS

### A.11.1 MATCHING THE NUMBER OF PARAMETERS BETWEEN RNN AND CV-RNN.

Table 5 showcases a significant difference between the number of parameters of the original and our CV-RNN. To ensure that the generalization abilities of the CV-RNN are not due to this increase of parameters but by the use of complex-valued units and the specific choice of operations to induce synchrony, we evaluate the performance of an RNN with more parameters. This new RNN (RNN-L – brown bar in Fig.A.12) is augmented by using a hidden dimension of 41 instead of 32. The resulting number of parameters adds up to $177,010$ (slightly more than the CV-RNN). However, the generalization abilities remain unchanged and still significantly lower than the CV-RNN.

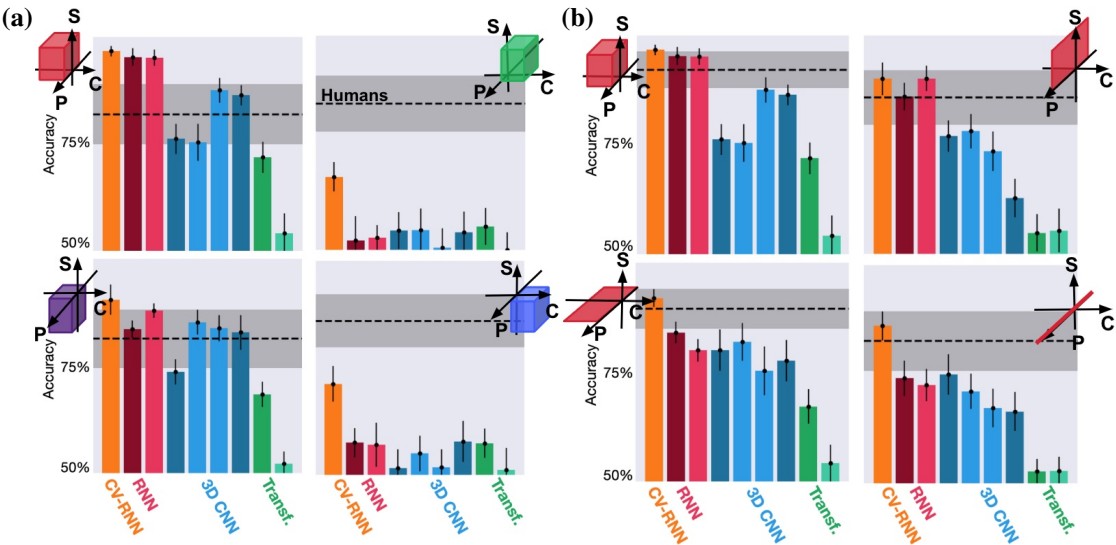

Figure A.12: Extended benchmark including an RNN (RNN-L) with the same number of parameters as CV-RNN (brown bar).

**Pre-training on all colors.** We hypothesize that the CV-RNN's failure to reach human performance in the OOD color conditions is due to a lack of prior knowledge of the entire colorspace. Because the models are trained on only half of the colorspace, the filters intended to represent the other half may not be initialized properly. Consequently, the model struggles to track the target under these conditions, not due to a failure of the circuit itself, but because the preprocessing layer (Conv3D in Table 4) does not provide accurate information to the circuit. To test this hypothesis, we train an RNN and a CV-RNN on a training distribution that includes objects exhibiting colors from the full colorspace and shapes evolving according to all the rules of the GoL combined. Once trained, we extract the weights of the preprocessing layer (Conv3D in Table 4), freeze them, and then train the RNN and CV-RNN circuits along with the classification layers. We then test the resulting models on all OOD conditions and compare them with other pre-trained models (pre-training on Kinetics400). We observe a significant improvement for both circuits (compared to the versions without pre-training), bringing the CV-RNN closer to human performance (see Fig. A.13).

We speculate that combining this pre-training with an advanced initialization (see Fig.A.5) could push the CV-RNN to achieve human-level performance in all conditions.

**Including Self-Supervised models.** We include in Fig. A.14 a self-supervised model, VideoMae (ViT-S with patch size 16) to evaluate whether the visual representation of such model can help solve `FeatureTracker`. We use pre-trained weights on Kinetics400 and finetune the model on the training set of `FeatureTracker`. The model's performance on the testing sets is very similar to the one of the supervised Transformers, suggesting a key role played by the architecture more than the training procedure to solve `FeatureTracker`.

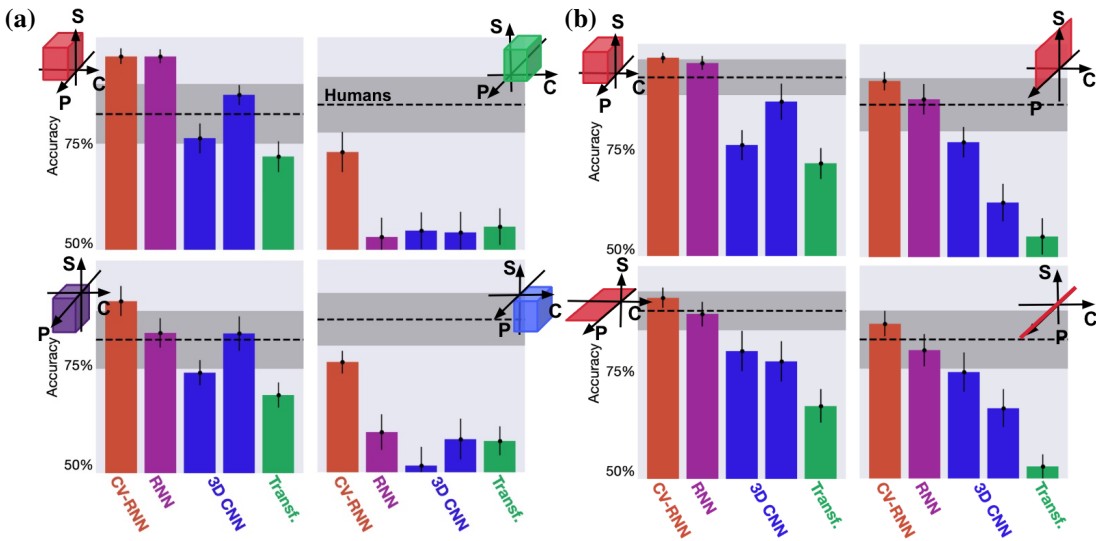

Figure A.13: Extended benchmark including models with pre-trained pre-processing layers on the full colorspace (RNN and CV-RNN) or Kinetics400 (3D CNN, Transformer).

We also trained DINO (Caron et al., 2021) on `FeatureTracker` to extract the features frame by frame (starting from pre-trained weights on ImageNet) and use an MLP or an RNN to perform the classification task. The model did not converge in training.

We further added results from the new model SAM2 (Ravi et al., 2024). Like DINO, this model is not designed for binary classification. Still, we can evaluate its ability to track the target across frames by comparing the position of the predicted mask with the actual position of the target. Specifically, we use the "tiny" pre-trained version of the model and initialize the first mask with the target's position. We evaluate the ability of the model to keep track of the target as it changes in appearance by defining the overall accuracy as the number of videos where the predicted mask was close to the actual position of the target (IoU > 0.9). We perform this evaluation on 1,000 images taken from the in-distribution test set. We report an accuracy of $0.001875\%(\pm 7.65e-05)$.

We did not include these two models in Fig. A.14 as the bars would all be at 50% in all the conditions.

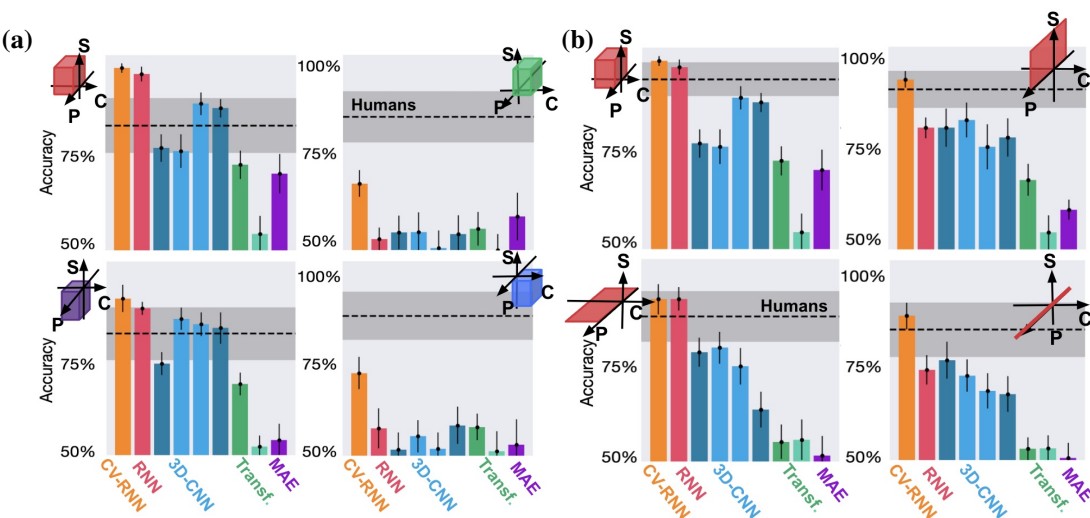

Figure A.14: Extended benchmark including a self-supervised model, Video-MAE (purple bar).

### A.12 ADDITIONAL RESULTS

#### A.12.1 VARYING THE TRAJECTORIES OF COLORS AND POSITIONS

Humans' tracking strategy suggests prioritizing the position feature over color and shape. In other words, humans track objects based on movement and rely on appearance only to disambiguate objects (e.g., in cases of occlusion). To confirm this hypothesis and evaluate if models employ a similar strategy, we test our observers on conditions where the color and/or spatial trajectories are more irregular than during training (see Sec. A.7 for details).

In Fig.A.15, we plot the difference between performance on predictable trajectories (i.e., a test set with a distribution similar to training) and performance on irregular trajectories, for both humans and models. A negative difference indicates a strong dependence on the feature whose trajectory is less predictable. Consistent with our predictions, humans show no difference in performance when the color becomes unreliable (top-left plot). However, performance decreases when the spatial trajectory is more erratic. Conversely, most models exhibit a significant drop in performance when the color trajectory is irregular, highlighting a strong dependence on color for task performance. They are also somewhat affected by changes in motion, though sometimes less so than by color changes. However, the CV-RNN displays behavior very similar to humans, being more affected by changes in motion trajectory than by changes in color trajectory.

Finally, the training dataset is generated in a way such that the target is never occluded by the distractor. Rather, it will cross the trajectory of some distractors but will always stay visible in every frame. We can consequently evaluate the ability of the models to generalize to a new condition where the target is occluded by the distractor when it crosses its trajectory. Fig. A.16 shows the performance on a set test set representing objects of the same feature statistics as the training distribution but where the target can be fully occluded by a distractor if it crosses its spatial trajectory. The CV-RNN remains much more robust than the baselines in this condition as well.

#### A.12.2 MORE NATURALISTIC CONDITIONS.

We create two new versions of the datasets with non-static backgrounds and textured objects as a first step towards more naturalistic stimuli.

To generate a non-static background, we model the background as a 3D Perlin noise, starting from a random state. In practice, the background is now colored (each sample starts from a random color and does not overlap with the colors of the objects) and evolves over time with a temporal and spatial dynamic different from the one of the objects. Considering the texture condition, we generate 5 possible textures (checkerboard, stripes, dots, noise, none). Each object is assigned a texture that will remain constant over time (only the noise texture changes over time). We control spatial, shape, and color dynamics similarly to the original version and report the accuracy of the CV-RNN and the baselines in Figures A.17 and A.18.

#### A.12.3 OBJECTS DISAPPEARING.

We finally evaluate whether the models can handle objects disappearing in the videos. To do that, we generate two new versions of the dataset, with objects moving in and out of the frame or containing objects that can vanish over time.

One version allows the objects to vanish by removing the last rule of the game of life (this rule being: "If no pixel is active, activate the pixel in the center"). The other version allows the objects to move along trajectories evolving in a window larger than the frame size. As a consequence, the objects can start their trajectories outside of the frame and appear in the frame at the middle of the video, or start inside the frame but move outside during the video and potentially reappear later. These new rules only apply to distractors and not to the target not to hamper the design of the task.

We first test the models on the respective test sets of these two versions using the models trained on the version where the objects never vanish or disappear. The results are shown in Fig. A.19, first row. While all the baselines show a clear drop in performance, the RNN and especially the CV-RNN's behavior remain very consistent across conditions. As a control, we train the models on the version of the dataset with objects moving in and out and test them on the other conditions (see Fig. A.19, second row). This new training condition improves the performance on the test set with objects

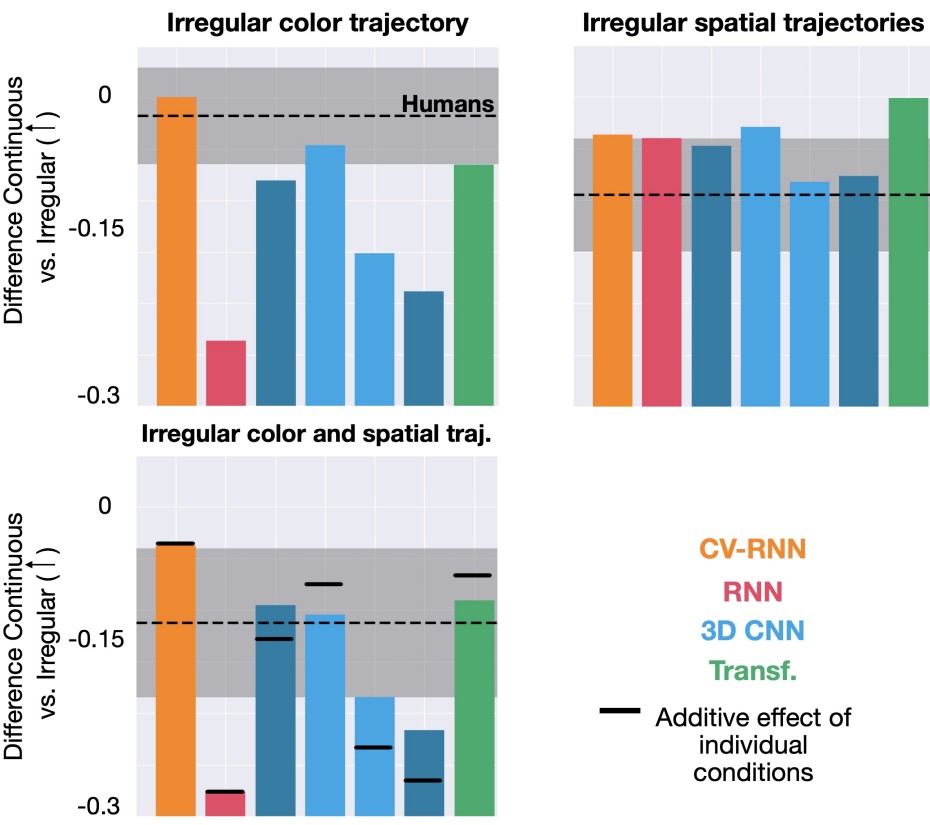

Figure A.15: The trajectories of colors and positions can be manipulated to highlight the tracking strategies of the models. In the top-left subplot, we report the difference in accuracy between test sets containing irregularly sampled colors and those with smooth trajectories from the training distribution. The top-right subplot shows the difference in accuracy between test sets containing spatial trajectories that are less smooth than those during training and test sets with identical distribution as during training. The bottom-left subplot represents the difference in accuracy between test sets with both irregular colors and positions and those with smooth trajectories from the training distribution. The horizontal black lines indicate the additive effect of both individual conditions. A value closer to zero indicates less dependence on the trajectory of the tested feature dimension.

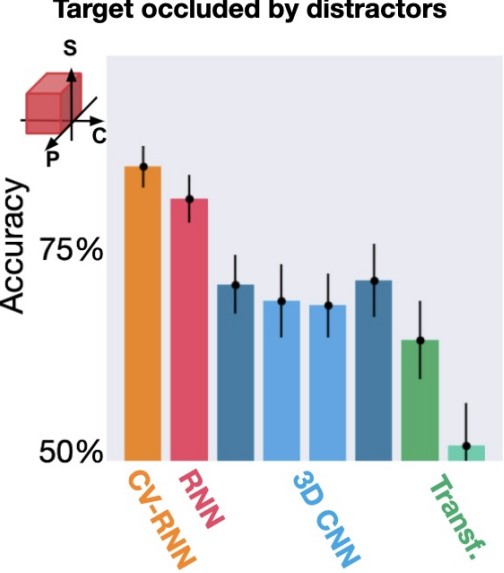

Figure A.16: Generalization ability of the models under conditions where the target is occluded by the distractor during a crossing.

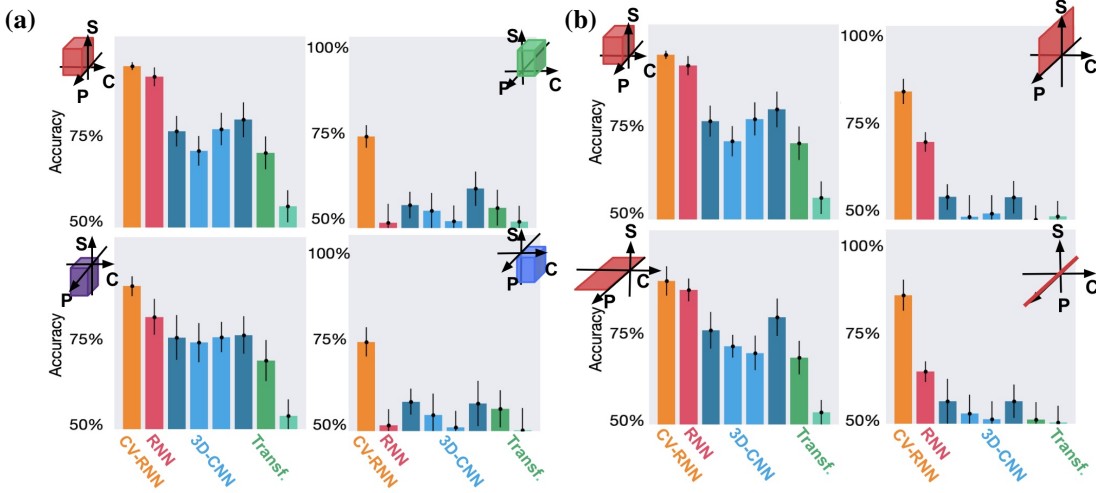

Figure A.17: Results of the CV-RNN (in orange) and the baselines on a version of `FeatureTracker` with non-static background.

moving in and out but also improves the robustness on the other conditions. However, the CV-RNN remains overall more accurate on all versions of the task.

## A.13 ERROR CONSISTENCIES

### A.13.1 COMPUTING ERROR CONSISTENCIES.

The "Error Consistency" measure (Geirhos et al., 2020b) quantifies the decision correlation (using Cohen's $\kappa$ coefficient) between two observers $i$ and $j$, corrected for accuracy. In practice, given $c_{obs_{i,j}} = \frac{e_{i,j}}{n}$ measuring the number of equal responses between both observers, the error consistency

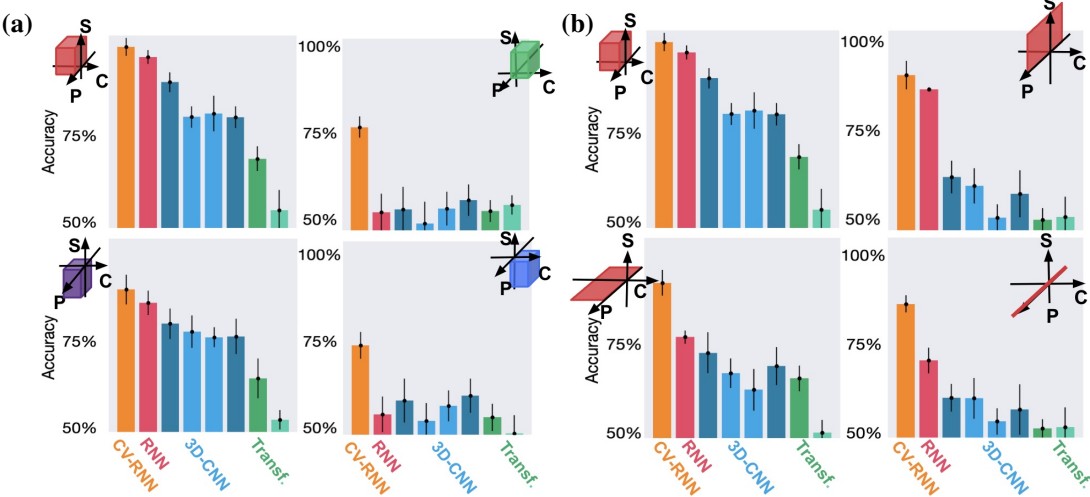

Figure A.18: Results of the CV-RNN (in orange) and the baselines on a version of `FeatureTracker` with textured objects.

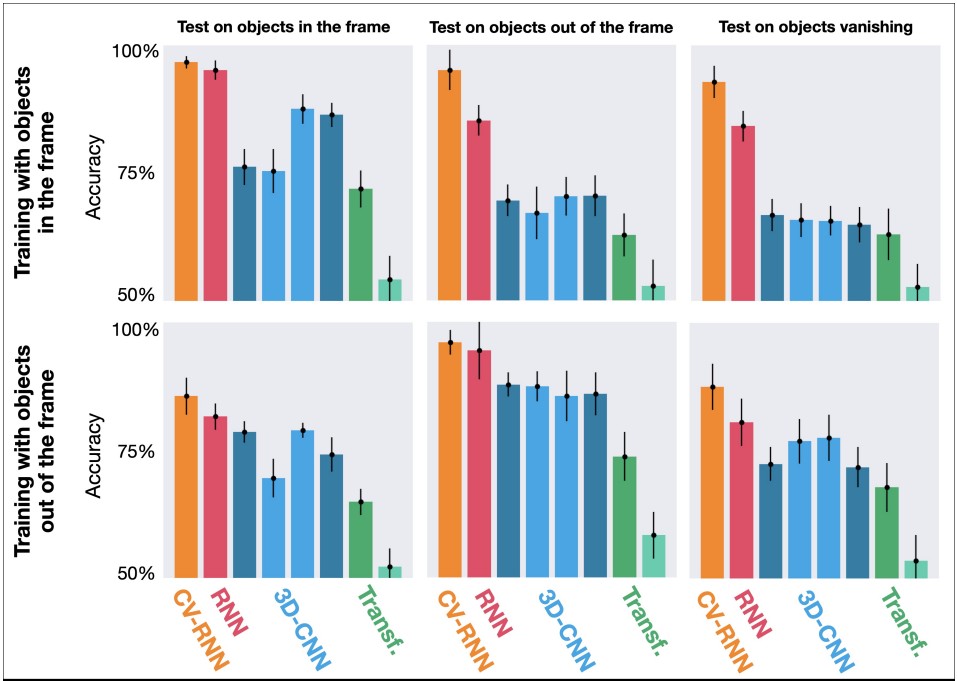

Figure A.19: The models learn the task where the videos contain a fixed number of objects always visible in the frames. We test their ability to keep track of the target when the distractors can move and out of the frame or vanish as their shape changes with time (first row). In the second row, we show the performance of models trained and tested with objects moving and out and tested on objects always in the frames or vanishing.

is computed with:

$$\kappa_{i,j} = \frac{c_{obs_{i,j}} - c_{exp_{i,j}}}{1 - c_{exp_{i,j}}} \tag{16}$$

where,

$$c_{exp_{i,j}} = p_i p_j + (1 - p_i)(1 - p_j) \tag{17}$$

measures the sum of the probabilities that two observers $i$ and $j$ with accuracies $p_i$ and $p_j$ will both give the same response, whether correct or incorrect, by chance.

### A.13.2 ERROR CONSISTENCY BETWEEN HUMANS AND MODELS.

Fig. A.20 shows the subject-to-subject Error Consistency. Overall, the level of agreement is positive and above 0.5, meaning that humans tend to make mistakes on the same videos. In Fig.A.21, we plot the model-to-subject Error Consistency. Compared to Fig.A.20, the score is now overall lower. This suggests a very different strategy between humans and models. However, the CV-RNN stands out by exhibiting a higher Error Consistency measure with human subjects.

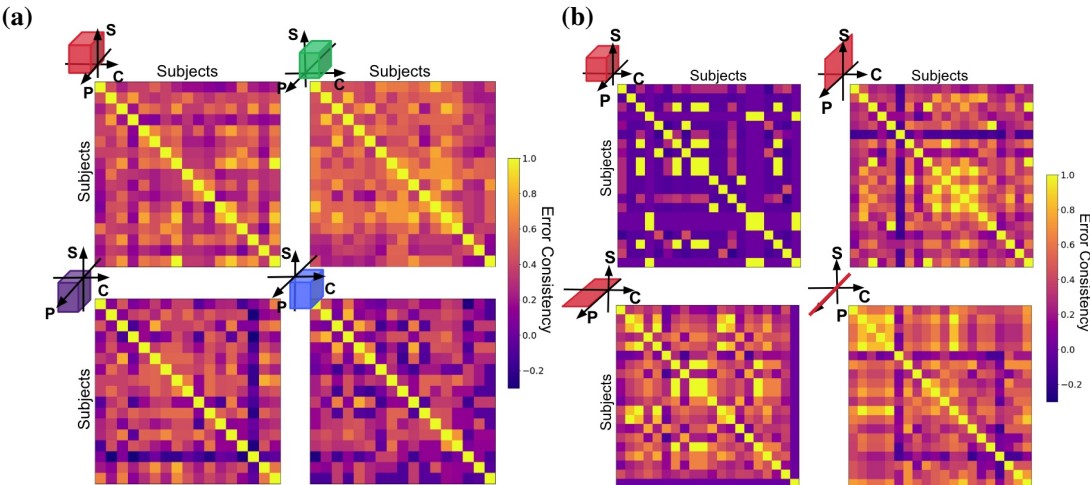

Figure A.20: The subject-to-subject Error Consistency measure represents the level of agreement between humans for each OOD condition. Each subject sees 30 videos from the test sets where features are out-of-distribution and 20 videos from the sets where the trajectories are fixed across time. For each condition, we report the Error Consistency measure between subjects computed based on the decisions taken for each video.

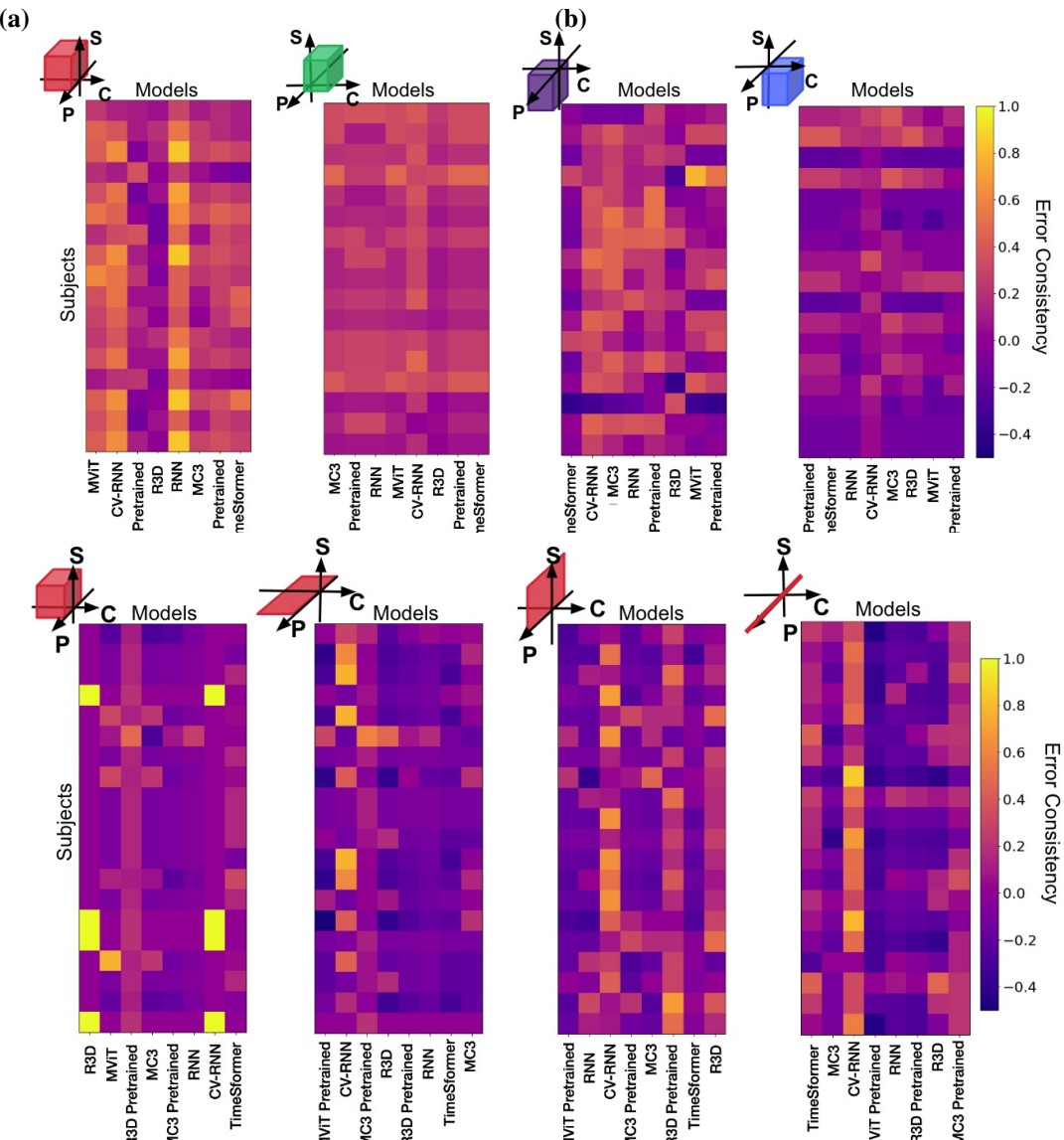

Figure A.21: The subject-to-model Error Consistency measure represents the level of agreement between humans and models for each OOD condition. Each subject sees 30 videos from the test sets where features are out-of-distribution and 20 videos from the sets where the trajectories are fixed across time. We present the same videos to the models and we report here the Error Consistency between subjects and models computed on the decisions taken for each video.

