# OpenReview forum: "Tracking objects that change in appearance with phase synchrony"
_ICLR.cc/2025/Conference — ICLR 2025 Poster_

### Official Review · Reviewer_vMMS · 2024-11-01

**Soundness:** 3
**Presentation:** 3
**Contribution:** 3
**Rating:** 8
**Confidence:** 4

**Summary:**

In this paper, the authors research how a neural network architecture and training scheme could be set up by drawing inspiration from neuroscience, such that it performs better in object tracking. In particular, in scenarios where the objects may change appearance. The main goal, however, is not simply increasing performance but rather to better understand observations in neuroscience by implementing hypothesized mechanisms in an ANN.
Concretely, in this task setting, object properties such as shape and color change while an object is moving over a trajectory, and the model has to decide if the selected object reaches the target position. Additionally, the object may be occasionally occluded. For training and benchmarking on this task, the authors construct a new dataset and challenge for object tracking. Human performance is assessed on the test set of this object tracking challenge. S.o.t.a. DNN models for object tracking are benchmarked and their failures analyzed. A new RNN architecture and training scheme with a specific loss function is established that leverages complex-valued attention units to simulate neural synchrony observed in real neurons of the visual pathway. The new architecture is able to reach human performance in several experimental settings and also shows similar failure modes as humans.

**Strengths:**

1. The paper establishes a new dataset and challenge for object tracking with changing object appearance. The setup to generate this synthetic data allows customization for specific research questions related to object tracking and ablation studies.
2. The authors present a new architecture and establish convincing experiments and results for why the new model (and training technique with specific loss function) brings the model mechanism closer to what is observed in neuroscience in terms of neural synchrony.
3. The paper is clearly structured and introduces the topic as well as explains the main goal in a clear fashion.  I like how the main model and challenge is introduced by first analyzing a more simple setting (shell game) both in humans and an initial model, which then leads to the main dataset and model. Most questions I had during reading were promptly answered in the main text or appendix, and the paper carefully sets expectations in terms of ML and neuroscience perspective, and the required ablations and add-on experiments were done.

**Weaknesses:**

1. The training and test data is made of synthetic data of single color shapes, it is unclear how well it translates to real videos (the authors leave this point for future work) and objects. An interesting intermediate step toward a more natural setting could have been textured shapes and/or a (ideally non-static) background image.
2. An interesting model to try besides the presented baselines would have been other complex-valued RNNs that are not bio-inspired but compatible with the introduced synchrony loss. This would have added the understanding of the impact of the bio-inspired architecture that InT provides.
3. Although different task variations are benchmarked, the limited scope to just one dataset (that the authors created) make it difficult to compare to other approaches that are specialized on changing appearance, e.g. [2] benchmarks on 4 datasets.

Just a suggestion (space is limited after all): there is a vast amount of literature of complex-valued RNNs (a survey that may also contain relevant RNNs is [1]) presented after the 'old' ones cited in the paper, and some citations could be added.

[1] Lee, ChiYan, Hideyuki Hasegawa, and Shangce Gao. "Complex-valued neural networks: A comprehensive survey." IEEE/CAA Journal of Automatica Sinica 9.8 (2022): 1406-1426.
[2] Cai, Yidong, et al. "Robust object modeling for visual tracking." Proceedings of the IEEE/CVF International Conference on Computer Vision. 2023.

**Questions:**

1. How do you assess the amount of error information that is provided during training of the baseline models vs the specific loss function (synchrony loss) for the CV-RNN? Could some of the increased test performance also be due to more efficient training with more loss information?
2. As far as I understand the data generation pipeline, all objects are initialized at the start and exist until the end of the sequence (besides occasional occlusions). Do you expect the model would still perform, if e.g. objects move in and out of the frame, or just vanish, or appear mid sequence? What if this only happens during testing?

---

> ### Author Response · Authors · 2024-11-22
>
> **The training and test data is made of synthetic data of single color shapes, it is unclear how well it translates to real videos…** Thank you for the suggestion. We agree that non-static backgrounds and textured objects would be a nice addition to the benchmark. We, therefore, generated two additional tasks, which we added to the challenge. Details on the generation of these new versions, as well as the latest results, are included in the manuscript. See Section A.12.2 and Figures A.17 and A.18. In short, the task seems harder for most of the models, but CV-RNN still outperforms the baselines on all conditions.
>
> **An interesting model to try besides the presented baselines would have been other complex-valued RNNs that are not bio-inspired but compatible with the introduced synchrony loss…** Very interesting point, thanks. We actually included an answer to this question in our original submission, but admittedly we were not thinking about it from the perspective of GRUs/LSTMs. In Fig A.5, we performed a wide set of ablations of our CV-RNN to better understand what parts were contributing to its success on FeatureTracker. One of these ablations involved removing the “excitatory” pool of neurons in the model, which essentially reduces the model to a version of a GRU with Neural Synchrony (as well as both additive and multiplicative interactions between neurons). This model performed very well overall but was a bit less robust than the full CV-RNN on unseen colors.
> Also, see Fig. A.5 for a version of the CV-RNN trained without the synchrony loss. This version rivals the RNN in all of the conditions, but performance is indeed boosted by the synchrony loss.
>
> **Although different task variations are benchmarked, the limited scope to just one dataset (that the authors created) make it difficult to compare to other approaches that are specialized on changing appearance, e.g. [2] benchmarks on 4 datasets.** Thanks for this comment. We have incorporated these two citations into our revision. See our **General Comments** for our thoughts on why we believe our focus on the Shell Game and FeatureTracker are significant for computational neuroscience, and validating the hypothesis that neural synchrony supports complex forms of object tracking in human vision.

---

> > ### Comment · Reviewer_vMMS · 2024-11-25
> >
> > Thank you to the authors for this detailed response and for considering the reviewer's input! The provided additional benchmarks and comparison to other approaches helped to better understand the approach and its performance.

---

### Official Review · Reviewer_oojY · 2024-11-03

**Soundness:** 2
**Presentation:** 2
**Contribution:** 2
**Rating:** 3
**Confidence:** 4

**Summary:**

The authors implement a particular biologically-inspired hypothesis for visual object tracking into a new-ish neural network (CV-RNN) type an then compare the results to standard network architectures.

**Strengths:**

I think the question of object tracking is interesting.

The benchmark is interestingly conceived.

**Weaknesses:**

The evaluations are very simple and don't really support strong claims about the new architecture being much better than previous architectures.

The test of "standard" DNNs for the purpose of baselines is perhaps a little shaky.

The benchmark, while interesting, is limited by being in such a toy condition.    I would be much more convinced if the algorithms here were also tested on recent object tracking benchmarks such as TAP-VID.  (Or if it was convincingly explained why such benchmarks are inapproprioate tests.)

**Questions:**

There are real-world object tracking benchmarks, such as TAP-VID.  Can the algorithms here work there?  Do they outperform  recent good strategies for tracking in those cases?

---

> ### Author Response · Authors · 2024-11-22
>
> **The evaluations are very simple…** Our synthetic datasets are visually simple but quite challenging for even state-of-the-art architectures to solve. This is because they focus on a specific visual challenge — the ability to track objects and maintain their identity even as they change in appearance over time. This challenge is a tightly controlled version of one we encounter everyday as we interact with the objects around us, for example when cooking. See our general comment for extended thoughts and how our approach is well-validated and standard in computational neuroscience.
>
> **The evaluations… don't really support strong claims about the new architecture being much better than previous architectures.** Our evaluations systematically probe models for their ability to track objects as they change in shape, color, shape and color, occlusion, etc. We believe that these evaluations are the most rigorous way to test what we set out to test: if neural synchrony could serve as a candidate mechanism for supporting robust object tracking. While computer vision benchmarks are more visually complex, they also open the door for models to rely on shortcuts that are not available in FeatureTracker to boost their performance (see related work section **Generalization and shortcut learning in deep neural networks** for more information).
>
> **The test of "standard" DNNs for the purpose of baselines is perhaps a little shaky.** As mentioned in our **General Comments**, We have expanded our benchmark to include VideoMAE, DINO, and SAM2. None of these models were successful on FeatureTracker. We are happy to add more models if the reviewer believes others might perform better — please let us know. We have also included a link to our code and data to support community efforts for benchmarking.
>
> **I would be much more convinced if the algorithms here were also tested on recent object tracking benchmarks such as TAP-VID. (Or if it was convincingly explained why such benchmarks are inapproprioate tests.)** We have now included VideoMAE, DINO, and SAM2 — state-of-the-art DNNs for visual tasks. These models as well as the other DNNs included in our original submission (R3D, MC3, TimeSformer, MViT – pre-trained or not) all struggle on FeatureTracker. As mentioned, we are happy to add more DNNs to the benchmark — we ask that the reviewer let us know which would be most impactful. We would also like to point out that while TAP-VID is a fascinating “dense point tracking” challenge, our focus was to systematically evaluate models and compare them to humans as fairly as possible. Having human responses on FeatureTracker is part of what makes it unique and significant.

---

> > ### Author Response · Authors · 2024-12-01
> >
> > As we approach the end of the discussion period, we wanted to ensure that all the reviewer's concerns or suggestions have been addressed. If there’s anything specific the reviewer would like us to elaborate on or discuss further, we’d be more than happy to do so.

---

### Official Review · Reviewer_EDdd · 2024-11-03

**Soundness:** 4
**Presentation:** 4
**Contribution:** 4
**Rating:** 8
**Confidence:** 4

**Summary:**

This paper presents CV-RNN, a deep learning circuit designed to replicate the biological visual system's ability to track objects with changing appearances by leveraging neural synchrony. The CV-RNN encodes object identity using the phase of complex-valued neurons, enabling the network to track objects as their appearance evolves over time. Through the "FeatureTracker" challenge, the authors demonstrate that while humans can effectively track objects despite changes in color and shape, conventional deep learning models struggle with this task. The CV-RNN, however, approaches human-level performance by employing phase synchronization to track objects with changing appearances.

**Strengths:**

1. The design of CV-RNN is well-motivated by prominent neuroscience theories (e.g., binding-by-synchrony) and serves as a proof-of-concept that neural synchrony aids in object tracking.
2. The combination of the FeatureTracker challenge and human psychophysics experiments provides a valuable framework for studying object tracking in a controlled environment, emphasizing the significant performance gap between current deep learning models and human capabilities.
3. CV-RNN outperforms other baseline models in the FeatureTracker challenge, demonstrating the efficacy of complex-valued representations.

Overall, the paper is well-motivated and clearly written. The main claims are supported by robust evidence.

**Weaknesses:**

1. Although CV-RNN approaches human performance on the FeatureTracker challenge, the use of synthetic datasets featuring simplified changes in appearance and shape may limit the generalizability of the findings to real-world object tracking, which is more challenging. Using naturalistic videos such as DAVIS would significantly strengthen the paper’s claims.
2. Comparing with self-supervised visual representation learning methods such as VideoMAE and DINO would be useful. The baseline methods in this paper — TimeSformer, MViT, ResNet3D, and MC3 — are supervised video action recognition models trained with specific activity labels, which may limit their generalizability to out-of-distribution datasets like the FeatureTracker challenge. Self-supervised learning methods have been shown to learn visual representations that transfer effectively to downstream vision tasks and could serve as competitive baselines for CV-RNN. If the authors believe self-supervised methods are not suitable baselines, it would be useful to provide a justification.

3. There is a missing comparison to RNN-L in Figure 5. The RNN performs worse than CV-RNN, which could be potentially due to its smaller number of parameters (108,214 vs. 171,580). The supplementary material (Table 5) mentions an RNN-L variant with a model size comparable to CV-RNN. Including its quantitative performance in Figure 5 would help demonstrate that the improvement of CV-RNN is independent of model size. If there was a specific reason for their omission, the authors can provide an explanation to clarify the rationale behind the omission.
4. Legend is missing in Figure A.10. Although efficiency is not the primary focus of this paper, an analysis of inference time and memory requirements for CV-RNN compared to baseline methods would be valuable. While Figure A.10 seems to include such a comparison, the missing legend makes it difficult to interpret.

**Questions:**

See Weaknesses

---

> ### Author Response · Authors · 2024-11-22
>
> **...the use of synthetic datasets featuring simplified changes in appearance and shape may limit the generalizability of the findings to real-world object tracking, which is more challenging…**: We addressed this point in our **General Comments** above. Our focus in this paper is to test the neuroscientific hypothesis of neural synchrony for tracking, which we found ample evidence for. We also wanted to emphasize that FeatureTracker is clearly challenging: while humans effortlessly learn an optimal strategy to solve the task despite being exposed to such videos for the first time, only the best models can reach human accuracy on the training distribution, and, except the CV-RNN, none of the models are able to learn an appearance-free tracking strategy necessary to solve the challenge. What makes FeatureTracker special is that we tightly control against shortcut learning, which means that models must learn a robust tracking strategy to solve the task.
> While we agree that it will be interesting to extend the CV-RNN to computer vision challenges like DAVIS, VOT, GOT-10K, etc., we believe that work is beyond the scope of our current manuscript, which already is 10 pages with 6 figures and an additional 25-page appendix with 21 figures.
>
> **[Comparisons] with self-supervised… methods…**  Thank you for these excellent suggestions. As discussed in our **General Comments**, we included three self-supervised models: VideoMae, the Segment Anything 2 model (SAM2) [1], and DINO [2].
> The VideoMAE failed to perform as well as the CV-RNN on FeatureTracker and was highly sensitive to the shapes of objects. The results of this experiment can be found in the Appendix (Section A.11.1 and Fig. A.14).
> SAM2 also struggled on FeatureTracker, and was outmatched by the CV-RNN. Because SAM2 is a segmentation model, we had to evaluate it slightly differently than the other classification models; we tested it by comparing the position of a predicted mask for the target object with the actual position of the target. Specifically, we use the "tiny" pre-trained version of the model and initialize the first mask with the target's position. We evaluate the ability of the model to keep track of the target as it changes in appearance by defining the overall accuracy as the number of videos where the predicted mask was close to the actual position of the target (IoU > 0.9). We perform this evaluation on 1,000 images taken from the in-distribution test set. We report an accuracy of 0.001875%
>  +/- 7.65e-05.
>
> Lastly, DINO performed at chance-level on FeatureTracker. In conclusion, the large-scale models in use today for many different types of image analysis tasks struggled on our FeatureTracker challenge. We believe this is because the specific tracking challenges introduced in FeatureTracker may require specific inductive biases, such as Neural Synchrony, that are currently not in the architectures or induced in the training routines of these models.
>
> [1] Ravi, Nikhila, et al. "Sam 2: Segment anything in images and videos." arXiv preprint arXiv:2408.00714 (2024).
>
> [2] Caron, Mathilde, et al. "Emerging properties in self-supervised vision transformers." Proceedings of the IEEE/CVF international conference on computer vision. 2021.
>
> **There is a missing comparison to RNN-L** This result is in Fig. A.12, which we did not include in the original Fig. 5 of the manuscript to limit visual clutter. It shows that the RNN cannot match the CV-RNN’s performance even after increasing its parameter count by 68,796. We can replace Fig. 5 with Fig. A.12 if the reviewer feels that this comparison should appear in the main text.
>
> **Legend is missing in Figure A.10…** We incorporated a legend in  Fig. A.10. Thanks for the suggestion. To summarize, the figure shows that the performance of the CV-RNN is not due to more parameters or flops than the baseline.

---

> > ### Comment · Reviewer_EDdd · 2024-11-26
> > **Thank you for the discussion!**
> >
> > Thank you to the authors for engaging in the discussion. I appreciate the time that put in to address my concerns and add comparisons to additional baselines. My concerns were sufficiently addressed by the response, so I raised my score. My primary concern was the use of FeatureTracker dataset, which can be very different from tracking in the real world. However, I agree with the authors that the dataset is sufficiently challenging to demonstrate the merit of CV-RNN, which is clearly better than baseline methods.

---

> > > ### Author Response · Authors · 2024-12-01
> > >
> > > We sincerely thank the reviewer for thoughtfully evaluating our rebuttal and for recognizing the improvements in our manuscript. The reviewer’s constructive feedback and positive reevaluation are greatly appreciated and have helped improve the quality of our work.

---

### Author Response · Authors · 2024-11-22
**General comments**

We thank the reviewers for their valuable feedback. The reviewers shared questions and concerns about (1) **Additional baseline models** , (2) **Translation** of our findings on synthetic datasets to computer vision benchmarks, which we will address here:

- **Additional baseline models**: Reviewers **EDdd**, **oojY**, and **vMMS** suggested we expand our FeatureTracker benchmark with self-supervised models (e.g., VideoMAE, DINO) and additional RNNs (e.g., non-bio-inspired complex-valued RNNs). We appreciate these insightful suggestions, and we have tested three new models on all versions of FeatureTracker: VideoMAE, DINO, and SAM2. The  CV-RNN outperformed each of these models on FeatureTracker, providing further evidence of the power of neural synchrony for tracking objects as they change appearances. We have included these results in our updated manuscript (see new Figure A.14).


- **Translation**: Reviewers **EDdd** and **oojY** asked for an extension of our CV-RNN to computer vision benchmarks. Our focus in this paper was to test the specific neuroscientific hypothesis that neural synchrony is a neurally plausible computational mechanism for tracking objects as they change in appearance. Questions about the role of neural synchrony in visual behavior have been debated in Neuroscience for decades, and by combining our CV-RNN with our two challenges (Shell Game and FeatureTracker), we were able to show that synchrony can support object tracking under challenging conditions, such as when objects change in appearance over time. While we agree that extensions to computer vision benchmarks would be interesting (also mentioned in the Discussion section L768), those experiments are beyond the scope of our computational neuroscience work which already has a main text with 10 pages with 6 figures and an additional 25-page appendix with 21 figures.
Note also that our focus on synthetic datasets to isolate and rigorously evaluate the ability of models to perform a specific visual computation follows a well-tread and powerful approach used by many others in computational neuroscience, which has been popular with the computer science community in conferences like ICLR and NeurIPS in the past (see related work section **Generalization and shortcut learning in deep neural networks** for other successful examples of using synthetic datasets for computational neuroscience).

To support future extensions of our CV-RNN to computer science benchmarks, we have released all of our model, experimental code, and data at https://anonymous.4open.science/r/feature_tracker-2CA3 (link included in the original submission).

We address additional points raised by the reviewers below, and we have uploaded a revision of our manuscript that incorporates all of the feedback. Our revision includes three new models (see new Figure A.14), model evaluations on four new versions of FeatureTracker that have more naturalistic visual elements (new Figures A.17, A.18 and A.19), and clarifications in the text based on the reviewers’ feedback.

---

### Meta-Review · Area_Chair_GKZe · 2024-12-20

**Metareview:**

This paper introduces a complex valued RNN designed to replicate the biological visual system's ability to track objects with changing appearances by leveraging neural synchrony. The authors also introduce a new “Feature Tracker” benchmark and show that  conventional deep learning models struggle to effectively track objects which change in color and/or shape, while humans succeed. Finally they demonstrate that  CV-RNN approaches human-level performance by using phase synchronization to track objects with changing appearances.
Strengths: All reviewers appreciated the newly introduced “Feature Tracker” benchmark. Two of the three reviewers also find the model well motivated by neuroscience, and consider the results a convincing proof-of-concept for the merit of synchrony in object tracking.
Weaknesses: The synthetic data of “Feature Tracker” are simplistic and results may thus not carry over to realistic cases.
While reviewer oojY is unconvinced by the papers results and significance, I agree with the other reviewers that this paper is an important contribution and recommend accepting it.

**Additional Comments On Reviewer Discussion:**

The review of reviewer oojY is overly brief and they did not react to the authors responses at all.
While their concerns are somewhat justified I don't think their assessment should carry a lot of weight.

---

### Decision · Program_Chairs · 2025-01-22

Accept (Poster)